# Underwater paleontology inside cenotes reveals the Miocene-Pliocene fish diversity in the Yucatan Peninsula, southeast Mexico

Kleyton M. Cantalice[1]*, Hugo E. Salgado-Garrido[2]☯, Erick Sosa-Rodríguez[3‡], Kay Vilchis-Zapata[4‡], Gerardo González-Barba[5]☯, on behalf of the Underwater Archaeological Atlas project[¶]

1 Departamento de Paleontología, Instituto de Geología, Universidad Nacional Autónoma de México, Ciudad de México, México, 2 Laboratorio de Carbonatos y Procesos Kársticos, Instituto de Geología, Universidad Nacional Autónoma de México, Ciudad de México, México, 3 Exploración Subterránea Responsable, Mérida, Yucatán, México, 4 Secretaria de Desarrollo Sustentable del Estado de Yucatán, Mérida, Yucatán, México, 5 Departamento Académico de Ciencias Marinas y Costeras, Universidad Autónoma de Baja California Sur, La Paz, Baja California Sur, México

☯ These authors contributed equally to this work.
‡ ESR and KVZ also contributed equally to this work.
¶ Membership of the Underwater Archaeological Atlas project is provided in the Acknowledgments.
* kleytonmc@geologia.unam.mx

## Abstract

The Yucatan Peninsula, southeast México, hosts a unique underwater karstic system of galleries connected by multiple sinkholes, locally called cenotes. This system is developed on a great Late Miocene to Early Pliocene carbonate platform belonging to the Carrillo Puerto Formation. The karstification process partially erodes these deposits' surfaces and exposes the fossil assemblage. Here, we present the fossil fish diversity in underwater prospections in the Cenotes Sambulá, San Juan, and X-Nabuy. Our results indicate the presence of at least 11 different taxa, which include: 1) species that live today on Mexican coasts, such as *Carcharhinus brachyurus*, *C. leucas*, *C. perezii*, *Carcharodon carcharias*, and unidentified *Rhinoptera* species; 2) extinct taxa, such as †*Galeocerdo mayumbensis*, †*Hemipristis serra*, and †*Otodus* (*Carcharocles*) *megalodon*; 3) taxa that are not currently distributed in Mexican coasts, such as *Carcharhinus macloti* and representatives of the genus *Aetomylaeus*. Furthermore, a new Diodontidae species, †*Chilomycterus dzonotensis* sp. nov., is described. It represents an increment in the Neogene fish diversity in the Gulf of Mexico and supports a shallow marine environment associated with a coral reef system. The small size of some teeth indicates that the deposits of Carrillo Puerto could be a shelter for tiny marine organisms, and the presence of some taxa highlights local extinctions in the Western Atlantic during the Late Cenozoic.

## Introduction

Cenote derives from the Maya word *ts'onot* or *d'zonot* to describe steep-walled natural wells generally opened by a cave roof collapse (collapse doline) or sinkholes. This structure provides

**Data Availability Statement:** All relevant data are within the manuscript and its Supporting Information files.

**Funding:** The work was performed through grant IA206123 provided by the Dirección General de Asuntos del Personal Académico (DGAPA)— Programa de Apoyo a Proyectos de Investigación e Innovación Tecnológica (PAPIIT). The funder's role in the study included specimen collection and data analysis.

**Competing interests:** The authors have declared that no competing interests exist

access to the water table and extends below it through a complex system of underground galleries [1]. In the Yucatan Peninsula (southeast Mexico), intense karstification processes have created one of the largest karst aquifers with many underwater and dry caves, including the world's longest underwater cave, the Sac Actun System in Quintana Roo state [2], and more than 3000 cenotes recorded in the state of Yucatan [3].

The Yucatan Peninsula's complex underwater caves and cenote systems are mainly developed in the Carrillo Puerto Formation sedimentary rocks, which range in age from the Late Miocene to the Early Pliocene [4]. Most of the Carrillo Puerto Formation evidence several fossil invertebrates, such as bivalves, gastropods, and corals, among other marine organisms [5–7]. Despite the abundance of ancient marine biodiversity on the sediments that compose the cenotes, the paleontological understanding of this geological formation is still developing since the fossil record remains almost entirely unidentified. This study presents the fish diversity from the Carrillo Puerto Formation obtained through underwater sampling in three distinct cenotes along the Yucatan State.

## Geological settings

The Carrillo Puerto Formation outcrops in the Yucatan Peninsula over an area of about 8800 Km2, which includes the central and eastern part of the state of Quintana Roo, the central region of Yucatán, and northeastern Campeche. It comprises limestones classified into mudstone, wackestone, grainstone, packstone, boundstone, and calcareous breccias, with thicknesses ranging from 5 to 25 meters [7].

The Carrillo Puerto Formation is divided into four distinct sections: 1) the base, formed by thick coquina strata covered with solid, white, or yellowish limestone, varying from cream-colored bioclastic packstone and calcarenites [5]; 2) the middle-level, conformed by limestones and yellowish and reddish clays, which by alteration give rise to red lateritic clays; 3) the upper level, represented by white, hard and massive limestone [6]; 4) the karst dissolutions and erosions of the lapiaz type observed in several representative outcrops of the unit [11, 12].

The Late Miocene to Early Pliocene age is attributed to the Carrillo Puerto Formation based on the presence of fossil foraminifera of the genera *Archaias* de Montfort, 1808, *Sorites* Ehrenberg, 1839, *Peneroplis* de Montfort, 1808 (*P. proteus* d'Orbigny, 1839), *Cycioloculina* Heron-Allen and Earland 1908, *Nonion* de Montfort, 1808, and *Gypsina* Carter, 1877, bivalves of the genera *Arca* Linnaeus, 1758 and *Venus* Linnaeus, 1758, turritellas, ostracods, and Hexacorallia corals [5, 11–13]. The Carrillo Puerto Formation correlates with La Laja, Depósito, and Encantado de la Cuenca formations in Veracruz State, Mexico. It overlies a small portion of the unidentified Oligocene formation and a significant portion of the Eocene Chichén Itzá Formation [11] (Fig 1).

Based on the lithological and faunal characteristics, the environment was interpreted as a shallow marine habitat corresponding to the neritic zone. These conditions persisted for a prolonged period, which must have covered a large part of the Peninsula where abundant fossils were developed in low-energy areas [7, 11]. The Elasmobranch and the teleost fossil remain herein described from the Carrillo Puerto Formation represent an increment in the knowledge about the marine diversity on the Gulf of Mexico during the Neogene and highlight local extinctions on the Western Atlantic after the Miocene-Pliocene boundary.

## Material and methods

A total of 22 fossil specimens were studied. Elasmobranchs and a teleost were collected in three cenotes (Sambulá, X-Nabuy, and San Juan) located in Yucatan state during the field trip in April 2023 (Fig 1A). The underwater collection used cave-diving vertical progression

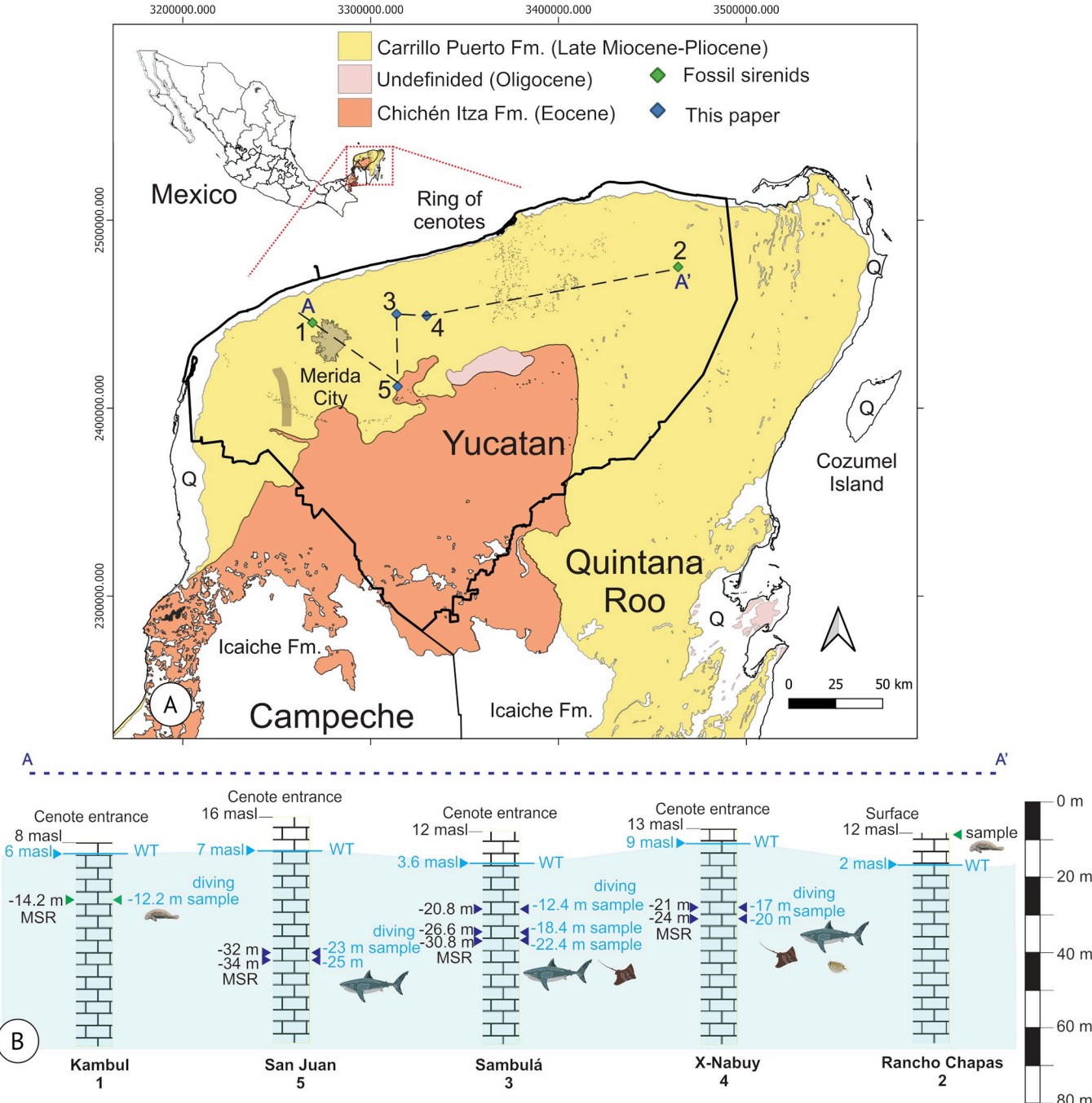

**Fig 1. Location map of fossil collection.** (A) Geological map with formations Icaiche, Chichén Itza, Carrillo Puerto, and Quaternary deposits (Q) modified from Servicio Geológico Mexicano (SGM) [8]. Paleontological records. 1 (Cenote Kambul) and 2 (Rancho Las Chapas) correspond to Sirenid for Domning [9, 10], while 3 Sambulá (Motul), 4 X-Nabuy (Suma), and 5 San Juan (Homún) cenotes correspond to this work. (B) Idealized stratigraphic unit of the Carrillo Puerto Formation. The cenote entrance is at a different altimetric position, meters above sea level (m.a.s.l.). Also, the water table (WT) starts at a different position. Diving starts at WT. Each cenote has a maximum depth at which the samples were collected. It corresponds to the maximum stratigraphic range (MSR), which includes the dive and meters above WT.

techniques and equipment, with Nitrox 32% tanks and side-mount configuration (Fig 2A, 2B, S1 File). The depth measures were taken during the diving using a dive computer, Teric Shearwater (Fig 2A). Two specialized divers conducted dives for collections for 60 minutes for each

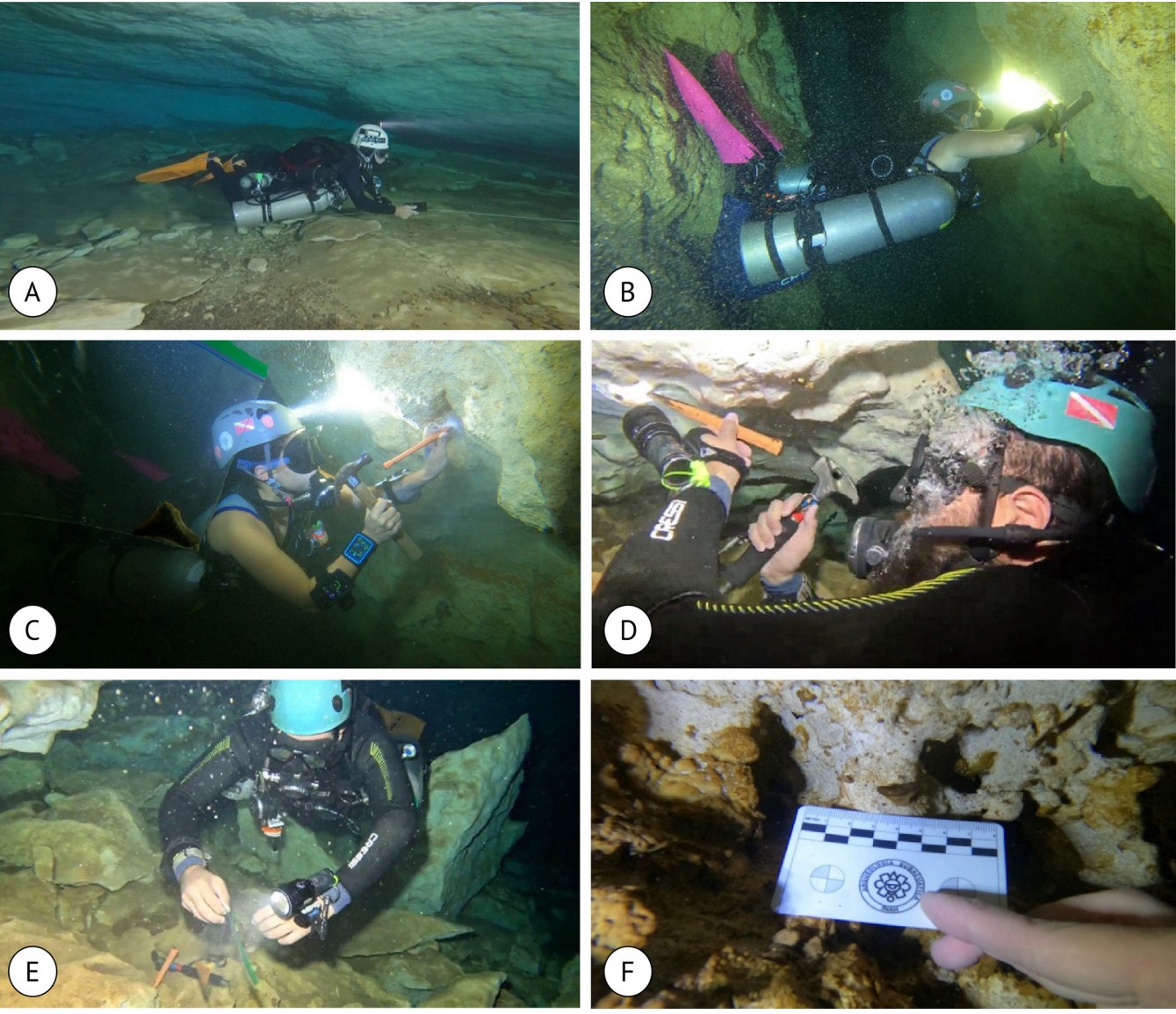

**Fig 2. Dive and sampling techniques inside cenotes.** (A) Side-mount diving technique. (B-E) Sample technique using chisel and hammer. (F) Tooth embedded in the wall.

person. The maximum depth reached was 25 meters in the Cenote San Juan (Fig 1B). In one of the cenotes, cave decompression techniques with multi-stations and rebreathers were also used, given the depth and bottom time. A chisel and hammer removed the specimens from the rock (Fig 2B–2F, S2 File). The samples were deposited in plastic bags to be transported to the surface and safeguarded in cotton wool.

The fossil material was cleaned using a micro jacket 2 tool and needles under a stereomicroscopic observation. In some cases, 0.5% sulfamic acid solution was directly applied to help the removal of well-consolidated sediments. Photographs were taken using a Nikon D5500 DSLR camera with an AF-S Micro-Nikkor 60mm f/2.8G ED macro lens. The figures were edited with the software Gimp v.2.10.34 and Inkscape v.1.2. The specimens are housed at the Colección Nacional de Paleontología of the Universidad Nacional Autónoma de México under the

acronym IGM. Elasmobranch anatomic interpretations follow that of Purdy et al. [14] (Fig 3), and porcupinefish osteology follows that of Tyler [15] and Aguilera et al. [16].

## Nomenclatural acts

The electronic edition of this article conforms to the requirements of the amended International Code of Zoological Nomenclature, and hence the new names contained herein are available under that Code from the electronic edition of this article. This published work and the nomenclatural acts it contains have been registered in ZooBank, the online registration system for the ICZN. The ZooBank LSIDs (Life Science Identifiers) can be resolved and the associated information viewed through any standard web browser by appending the LSID to the prefix ""http://zoobank.org/"". The LSID for this publication is: urn:lsid:zoobank.org:pub: 114D2528-7A15-4318-9932-0B584463D934. The electronic edition of this work was published in a journal with an ISSN, and has been archived and is available from the following digital repositories: PubMed Central, LOCKSS.

## Abbreviations

**IGM.** *Instituto Geológico de México* [Mexican Institute of Geology], is the catalog registration abbreviation for specimens deposited in the Colección Nacional de Paleontología [National Collection of Paleontology of Mexico]. IGM-loc. is the catalog abbreviation for fossil localities on the National Collection of Paleontology of Mexico. **INAH**, *Instituto Nacional de Antropología e Historia* [National Institute of Anthropology and History]. **UNAM**, *Universidad Nacional Autónoma de México* [Autonomous National University of Mexico].

# Results

## Systematic paleontology

Phylum Chordata
 Infraclass Elasmobranchii Bonaparte, 1838 [17]
 Division Selachii Cope, 1871 [18]
 Order Carcharhiniformes Compagno, 1973 [19]
 Family Carcharhinidae Jordan and Evermann, 1886 [20]
 Genus *Carcharhinus* Blainville, 1816 [21]
 *Carcharhinus brachyurus* Günther, 1870 [22]
 (Fig 4)

## Referred material

IGM 13913 is a complete tooth with the crown turned to the right. IGM 13914 is a small tooth with the crown turned to the right, and the root is eroded. IGM 13915 is a complete tooth with the crown turned to the left. IGM 13916 is a tooth with the crown turned to the left and a small part of the distal root lobe absent. All the specimens were collected in the cenote X-Nabuy (Fig 1).

## Description

Four teeth from 6.7 to 15.2 millimeters (mm) in total length and from 6.1 to 11.2 mm in height (S1 Table). The teeth are flattened labio-lingually and have a narrow triangular crown. Lingually, the crown represents from 60.9 to 75.9% of the total height, and the root comprises 35.6 to 47% of the total height. The crown is triangular and narrow, with the mesial and distal cutting edges slightly curved and serrated. In both mesial and distal views, the labial apex projection and the sigmoid lingual edge are pronounced in IGM 13913 and IGM 13916 (Fig 4). A

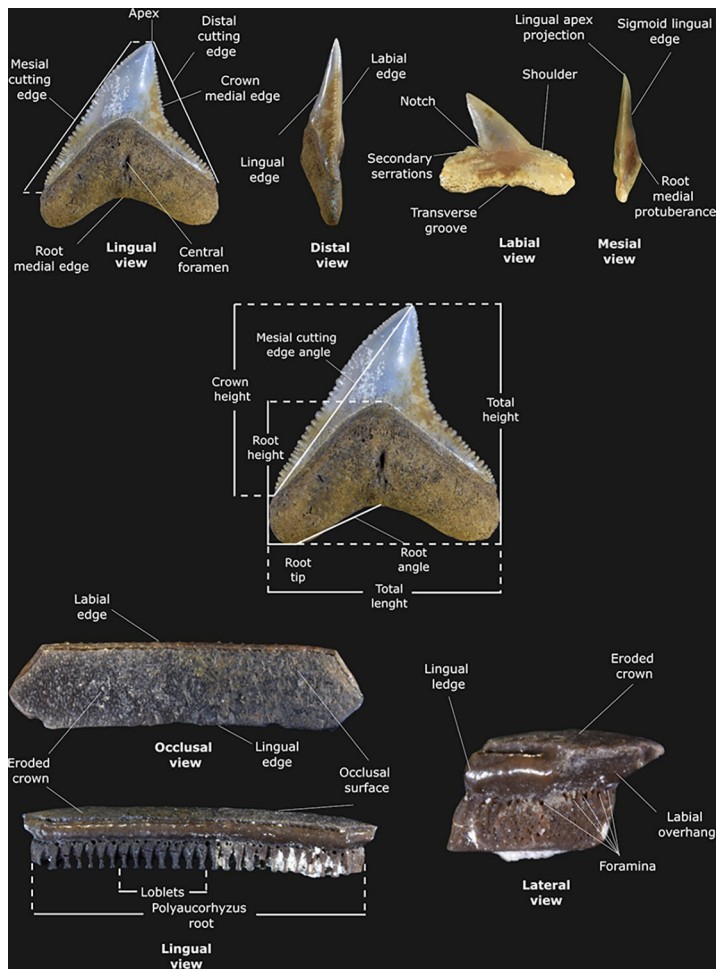

**Fig 3. Measures performed on the elasmobranch specimens studied here.** The specimens IGM 13917, IGM13921, and IGM 13931 were used as models.

deep and acute notch separates the crown and shoulder on the distal surface. The shoulders are laterally large and enameled. The serrations are vestigial in the apical portion of the crown, becoming more developed over the shoulders and coarse on the distal cutting edge in IGM 13914 and IGM 13915. Secondary serrations and cusplet are lacking. The crown medial edge is not deep. The mesial cutting edge creates an angle ranging from 38.2° to 43°. The ventral edge of the root is also shallow. The root angle ranges from 13.6° to 18.8°. The root medial protuberance is well-developed. The central foramen is housed in a deep, transverse groove. The transverse groove is notable in labial and lingual views (Fig 4).

## Remarks

The diagnostic features that indicate these specimens as *Carcharhinus brachyurus* (the copper shark) include the narrow triangular, strongly serrated, and oblique crown. Furthermore, the curvature of the distal cutting edge forms a deep acute angle, separating the serration of the shoulders from the apex of the crown [14, 23].

*Carcharhinus brachyurus* is distributed in other Miocene-Pliocene fossil localities from Brazil, Colombia, Venezuela, USA, Portugal, and France [14, 24–26]. In Mexico, specimens of *C*.

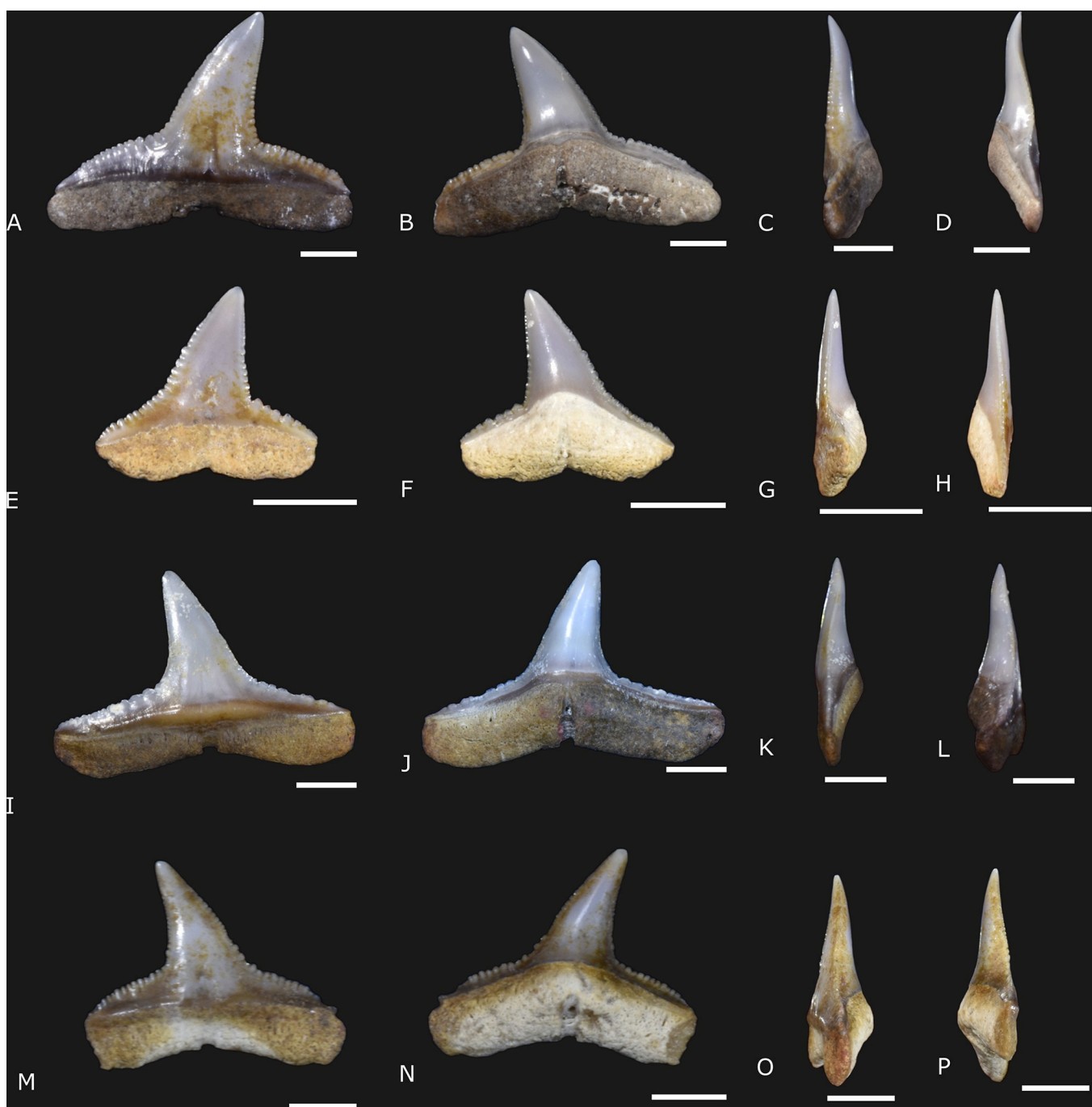

**Fig 4. *Carcharhinus brachyurus* Günther, 1870 [22] specimens (IGM 13913 to IGM 13915, and IGM 13916) from the Carrillo Puerto Formation.** (A, E, I, and M) Labial view. (B, F, J, and N) Lingual view. (C, G, K, and O) Right lateral view. (D, H, L, and P) Left lateral view. The scale bars indicate 3 mm.

*brachyurus* are reported from the Miocene-Pliocene strata of Baja California [27–29]. Living specimens have a worldwide distribution, inhabiting coastal environments on both sides of Mexico. Extant *C. brachyurus* reaches up to 320 m in marine environments and is associated with reef systems, estuaries, and occasionally in freshwater [30].

*Carcharhinus leucas* Valenciennes, 1839 [31]

(Fig 5)

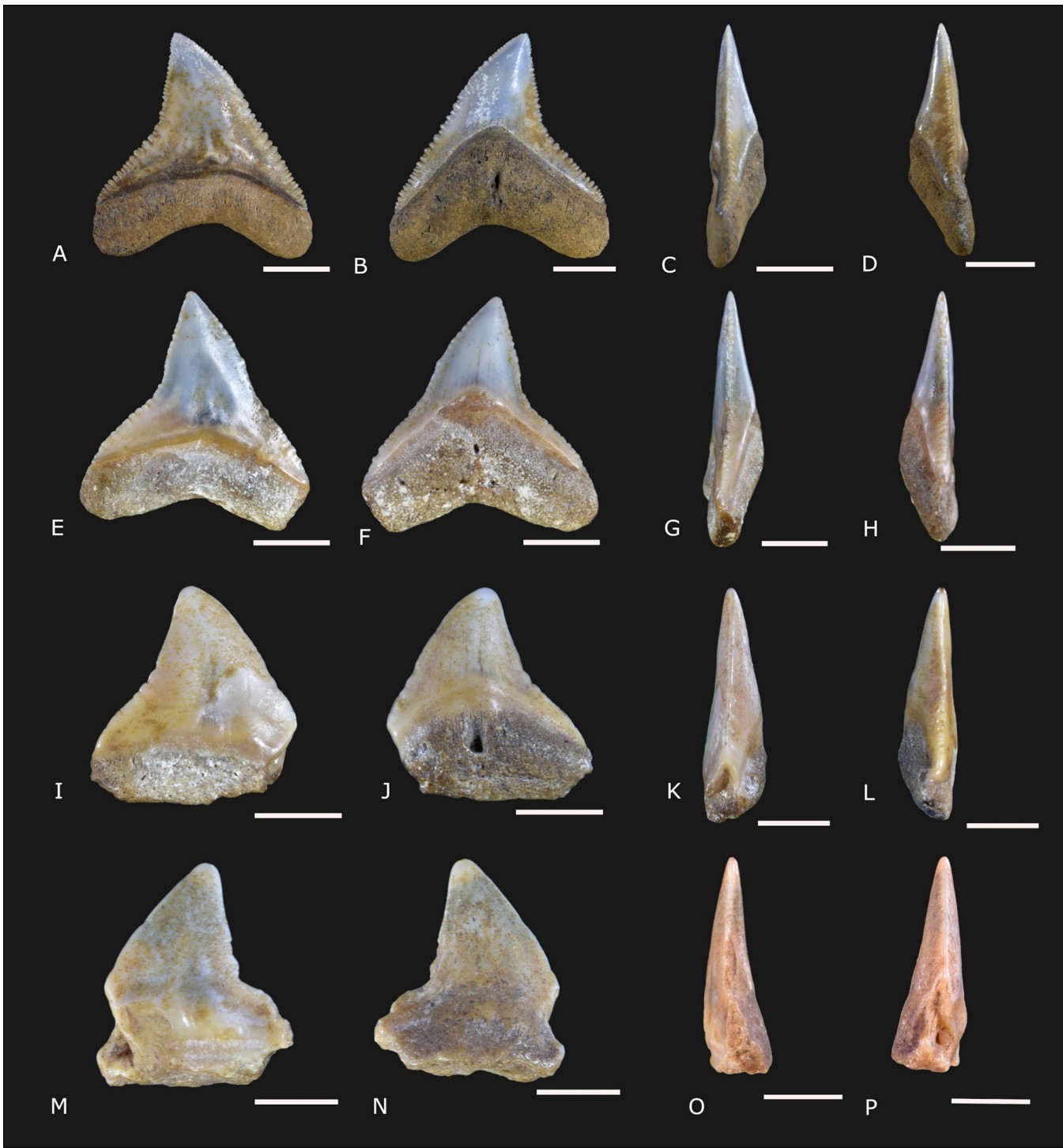

**Fig 5. *Carcharhinus leucas* Valenciennes, 1839 [31] specimens (IGM 13917 to IGM 13920) from the Carrillo Puerto Formation.** (A, E, I, and M) Labial view. (B, F, J, and N) Lingual view. (C, G, K, and O) Right lateral view. (D, H, L, and P) Left lateral view. The scale indicates 5 mm.

## Referred material

IGM 13917 is a complete tooth with the crown turned to the left. IGM 13918 is an almost complete tooth with the crown turned to the left, the cutting edges are damaged, and a small portion of the root lobe is lost. IGM 13919 is an eroded tooth with the crown turned to the left, the incomplete root, and vestigial cutting edges. IGM 13920 is a very eroded crown, turned to the right. All the specimens were collected in the cenote X-Nabuy (Fig 1).

## Description

Four teeth range from 15.5 mm to 17.5 mm long and 15.4 mm to 18.3 mm in height in complete specimens (IGM 13917 and IGM 13918) (S1 Table). The teeth are labio-lingually flattened with a large triangular crown representing 77.1 to 79.1% of the total height, while the root represents 59.1 to 61.2% of the total height in lingual view. Both labial and lingual surfaces are straight. The lingual surface is inclined to the distal edge, and the apex is not labially curved (Fig 5). A medial constriction on the medial portion of the crown produces a curvature in both cutting edges, which is more pronounced on the distal edge. The mesial cutting edge produces an angle ranging from 35 to 37.3˚. Both cutting edges are serrated. The serrations are fine close to the apex and become coarse in the base of the crown (Fig 5). Cusplets are lacking. The crown ventral edge is arcuate. The root presents a deep curvature on the ventral edge. The root angle observed represents 22.9 to 24.7˚. The root medial protuberance is not strongly pronounced. The central foramen is present, and the transverse groove is lacking (Fig 5).

## Remarks

The tooth is attributed to *Carcharhinus leucas* (the bull shark) due to its labiolingually flattened condition, with a broad triangular crown and an oblique apex. The cutting edges are shallow and notched on both sides but are more developed on the distal surface. Serrations along both cutting edges are more developed on the crown base and the shoulders. The root is arcuate, showing a central foramen and no transverse groove [14, 32].

The fossil record of *Carcharhinus leucas* dates to the Miocene and is distributed in outcrops from Brazil, Egypt, India, Panama, Perú, Portugal, Italy, Angola, USA, among others, e.g., [33–35]. The first Mexican *C. leucas* fossil record is from Miocene-Pliocene outcrops of Baja California [27, 28]. This species currently inhabits shallow marine waters of less than 30 m depth and occasionally migrates to freshwater and deeper waters close to 150 m [23].

*Carcharhinus macloti* Müller & Henle, 1839 [36]

(Fig 6A–6D)

## Referred material

IGM 13921 is a complete tooth with the crown turned to the right (Fig 6A–6D). The specimen was collected in the cenote X-Nabuy (Fig 1).

## Description

There is a single specimen with 7.07 mm in total length and 5.1 mm in height (Fig 6A–6D). The tooth is flattened labio-lingually, with the maximum height of the crown representing 70% of the total height of the tooth in the lingual view. The root occupies 49% of the total height in the same view. The crown is narrow and triangular, with the apex turned to the left in labial view (Fig 6C, 6D). The labial surface of the crown is straight, while the lingual surface is medially concave, forming a sigmoid curvature observed in both distal and mesial views. The apex is slightly curved to the labial portion. The mesial cutting edge is straight and

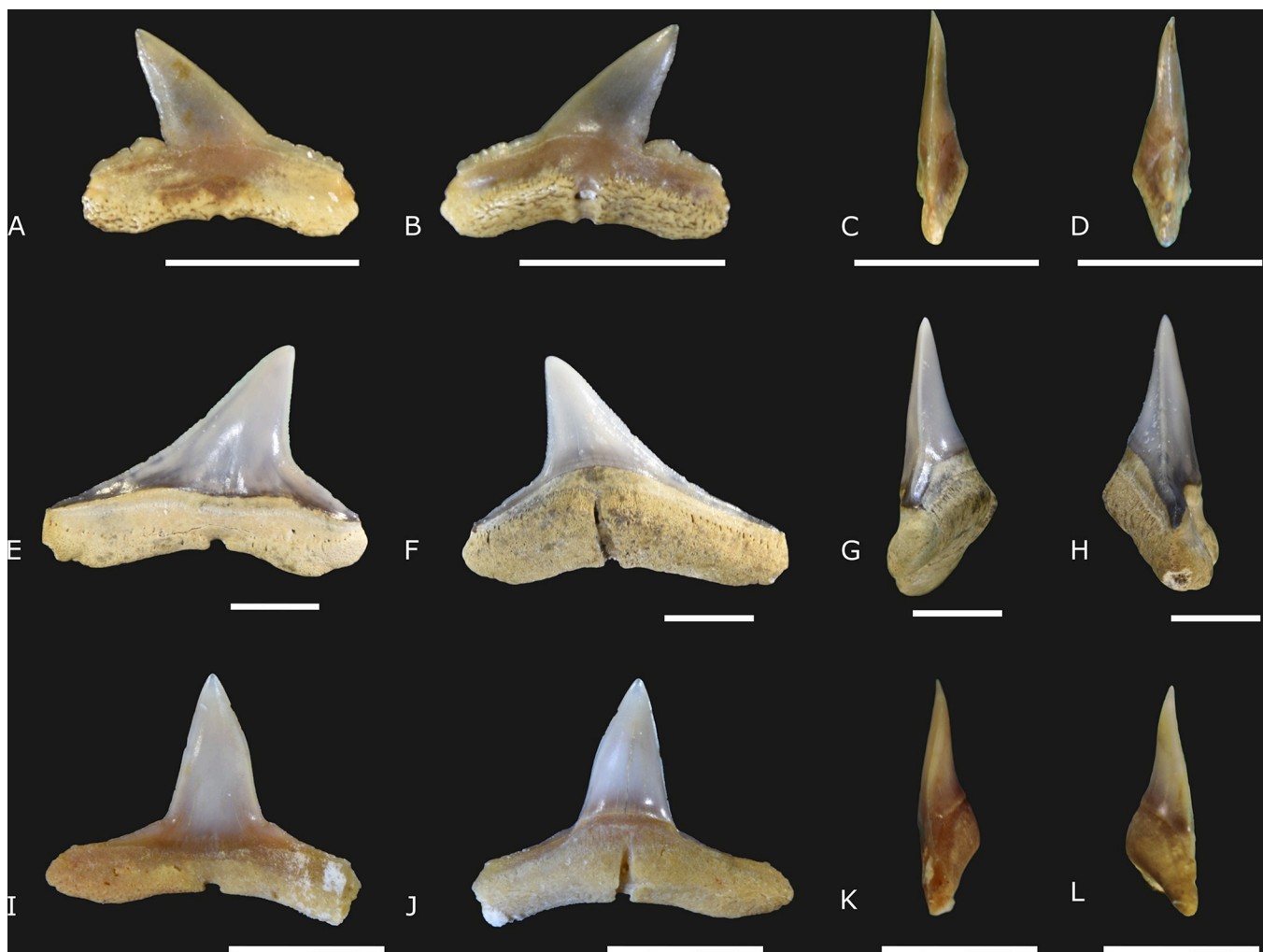

**Fig 6. *Carcharhinus macloti* Müller & Henle, 1839 [36] (IGM 13921), *C. perezii* Poey, 1876 [37] (IGM 13922), and *Carcharhinus* sp. (IGM 13923) from Carrillo Puerto Formation.** (A, E, and I) Labial view. (B, F, and J) Lingual view. (C, G, and K) Right lateral view. (D, H, and L) Left lateral view. The scale indicates 5 mm.

inclined, forming an angle of 34° around the apex. The distal cutting edge presents a deep and acute notch. Both mesial and distal cutting edges are smooth. Cusplets are lacking. The shoulders are large and enameled. The dorsal surface of the mesial shoulder is adorned with small and weak serrations on the mesial edge. The serrations on the distal edge shoulder are more developed and present a small cusplet-like shape. The crown ventral edge is arcuate. The root is also poorly arcuate and shallow. The root angle in this specimen forms an angle of about 11°. The root lingual protuberance is triangular and pronounced. The base of the root lobes is light cream, and the external surface is ridged, mainly on the lingual view. Both the central foramen and transverse groove are well-developed. The transverse groove is visible on both lingual and labial views.

## Remarks

The single specimen herein studied is attributed to *Carcharhinus macloti* (the hardnose shark) due to the small tooth with a narrow crown and smooth cutting edges in the apical portion of the crown and developed serrations on the shoulders, showing a cusplet-like shape on the

distal surface [14, 33]. *Carcharhinus macloti* is like *Carcharhinus dilcemai* [38] from the Early Miocene of the East Pisco Basin, Chilcatay Formation, Peru [38]. The species herein described is more related to *C. macloti* due to the absence of twist in the crown, a diagnostic feature found in *C. dilcemai* [38].

The fossil record of *C. macloti* is scarce compared with other requiem sharks. *Carcharhinus macloti* teeth were previously reported from the Early Miocene Caribbean seas of northeastern Amazonian, in Brazil, on Miocene Western North Atlantic marine outcrops from Maryland, and the Miocene-Pliocene from North Carolina [14, 33, 39]. The fossil herein described is the first report from Miocene-Pliocene marine fossil beds from the Gulf of Mexico, indicating a wide distribution of *C. macloti* in the Western Atlantic coasts. Living hardnose sharks are not distributed in the Atlantic but are restricted to continental and insular shelf environments from the Indo-West Pacific [33, 40].

*Carcharhinus perezii* Poey, 1876 [37]

(Fig 6E–6H)

## Referred material

IGM 13922 is a well-preserved tooth with a crown on the left side. Only a tiny portion of the external border of the mesial root lobe is lacking. The specimen was collected in the cenote X-Nabuy (Fig 1).

## Description

There is a unique labio-lingually flattened tooth with 18.27 mm of total length and 12.83 mm of total height (S1 Table). The height of the crown represents about 76% of the total height in the lingual view. The highest portion of the root represents about 52% of the total height. The enamel is well-conserved on the labial and lingual surfaces of the crown. The crown is triangular and narrow, with the apex directed to the right side on the labial view. The labial surface is almost straight, and the apex is not projected in lateral view. Both mesial and distal cutting edges are finely serrated (Fig 6E, 6F). The serrations around the apex are minute. The mesial cutting edge forms an angle of 37.2° around the apex. The notch in the distal surface is not deep, and the cusplets are lacking. The shoulder is large and enameled. The root has a very pronounced lingual protrusion, with a well-defined central foramen embedded in a deep, transverse groove visible in both labial and lingual views. The basal margin of the root is slightly curved. The root creates an angle of about 10°.

## Remarks

This single specimen is attributed to *C. perezii* (the Caribbean reef shark) due to the presence of diagnostic features. These include a large and triangular crown with vestigial serrations along both the cutting edges and the curvature of the cutting edges not pronounced [14, 32, 33]. The fossil record of *C. perezii* is reported in Miocene-Pliocene outcrops from the USA, Panama, Portugal, Italy, and others [14, 32, 33, 41]. It is the first fossil record of *C. perezii* from Mexico. Extant species are distributed in the western Atlantic, including the Gulf of Mexico and the Antilles [23, 40]. *Carcharhinus perezii* is a reef-associated species, usually ranging from 1 to 35 meters [14, 23].

*Carcharhinus* sp.

(Fig 6I–6L)

## Referred material

IGM 13923 is an almost complete tooth with only a tiny portion of the mesial root lobe missing. The specimen was collected in the cenote San Juan (Fig 1).

## Description

There is a single specimen with 10.45 mm in total length and 7.81 mm in height (S1 Table). The tooth is flattened labio-lingually and has a straight and triangular crown, with about 77% of the total height and the root occupying 43% of the total height in lingual view. The tooth crown is triangular and narrow, with the apex pointing upwards. Both labial and lingual surfaces are curved, and the apex is labially curved, forming a lateral sigmoid curvature in lateral view. The mesial cutting edge is slightly convex (Fig 6I and 6J), creating an angle of 39.6° with the apex. The distal cutting edge is straight and somewhat inclined. Both mesial and distal cutting edges are smooth. The curvature of the medial portion of the ventral border of the crown is not very pronounced. The shoulders are broad, and the enameled portion over its dorsal surface is thin. There are no serrations or cusplets over the shoulders. The root is not deep, but the lobes are large. The curvature in the ventral edge of the root is less pronounced. The root angle is 14.5°. The root medial protuberance in the lingual surface is almost triangular and very pronounced. The central foramen is well-developed and embedded in a deep transverse groove visible in both labial and lingual views.

## Remarks

The specimen is not attributed to any known species since only general features of *Carcharhinus* (requiem sharks) are preserved. The features include the semi-erect crown with a smooth cutting edge, enameled and large shoulders, and a large root with the central foramen and notable transverse grooves. According to Purdy [32], these features are related to the lower jaw tooth of a general *Carcharhinus* species and, therefore, could belong to any of the species described above. For this reason, we prefer to maintain this specimen as a not-determined member of the genus *Carcharhinus*.

Genus *Galeocerdo* Müller & Henle 1837 [42]

†*Galeocerdo mayumbensis* Dartevelle & Casier, 1943 [43]

(Fig 7A–7D)

## Referred material

IMG 13924 is an incomplete tooth with the crown turned to the right. The root lobe and the basal portion of the crown on the distal edge are lacking. The specimen was collected in the cenote Sambulá (Fig 1).

## Description

There is a single specimen with about 17.19 mm in total length and 24.86.3 mm in height (S1 Table). The tooth is labio-lingually flattened. The crown height is about 74.7% of the total height in the lingual view. The highest portion of the root represents about 53% of the total height at the same plan. The crown is triangular and large, with the apex directed to the right in labial view (Fig 7A and 7B). In both mesial and distal views, the labial surface of the crown is concave, while the lingual is convex. The apex in †*Galeocerdo mayumbensis* is directed to the labial portion (Fig 7C and 7D) but does not have the lingual sigmoid curvature, such as is found in *Carcharhinus* sp. for example (Fig 6K–6L). The mesial cutting edge creates an angle of 27.4°. The notch in the distal cutting edge is acute and shallow. The cutting-edge serration in the apical portion of the crown is thin. However, at the base of the crown, the serrations become coarse and triangular, showing secondary serrations (Fig 7A and 7B). Both coarse and secondary serrations are more developed on the basal portion of the distal cutting edge. Cusplets are lacking. The ventral border of the crown is arcuate. The root angle in this specimen

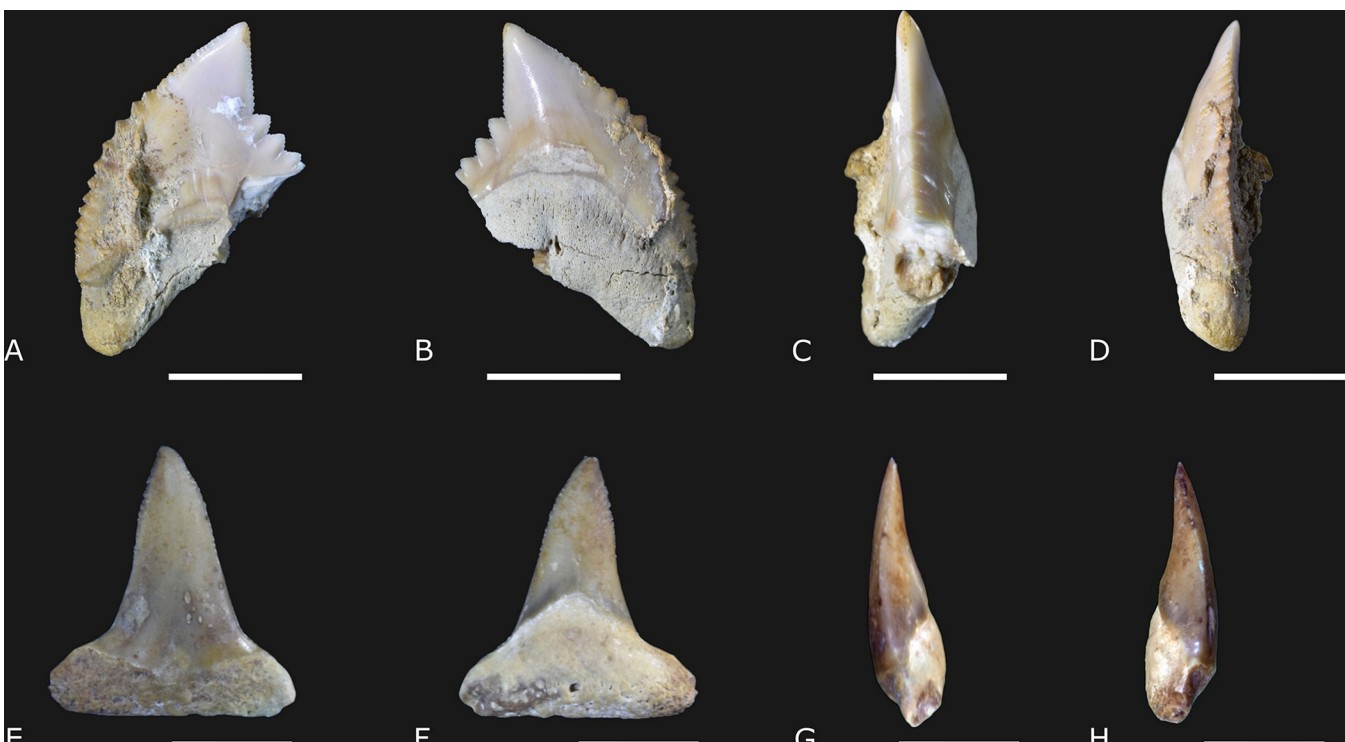

**Fig 7.** †*Galeocerdo mayumbensis* Dartevelle & Casier, 1943 [43] (IGM 13924; A-D) and †*Hemipristis serra* Agassiz, 1838 [53] (IGM 13925; E-F) from Carrillo Puerto Formation. (A and E) Labial view. (B and F) Lingual view. (C and G) Right lateral view. (D and H) Left lateral view. The scale bar from (A) to (D) represents 10 mm, while from (E) to (H) comprises 5 mm.

represents about 30.1˚. The root ventral edge is deep and arcuate. The root lingual protuberance is semicircular and less pronounced. The central foramen is present, and the transverse groove is lacking.

## Remarks

The specimen studied here is attributed to the extinct tiger shark fossil species, †*Galeocerdo mayumbensis*, and shows several diagnostic features. These include a broad triangular crown with an oblique apex, the lesser concave distal cutting edge, the presence of coarse serrations with secondary serration on the distal cutting edge, and a deep root with a solid arcuate ventral edge, and without a transverse groove [14, 44, 45].

†*Galeocerdo mayumbensis* was initially described from the Miocene beds of Western Africa. However, the current distribution includes Miocene outcrops from Brazil, Colombia, Madagascar, and Libya [16, 24, 45, 46]. It is assumed that some *Galeocerdo* specimens found in Miocene outcrops from the USA, Panama, Brazil, and India could be synonyms of †*G. mayumbensis* [24]. Nevertheless, revising these materials is still necessary to corroborate this argument. The specimen described herein is the first evidence of †*G. mayumbensis* in outcrops from the Western North Atlantic, confirming the hypothesis that this species was widely distributed in shallow marine environments during the Miocene [24].

Family Hemigaleidae Hasse, 1879 [47]

Genus *Hemipristis* Agassiz, 1835 [48]

†*Hemipristis serra* Agassiz, 1835 [48]

(Fig 7E–7H)

## Referred specimen

IGM 13925 is an incomplete lower tooth with the root partially preserved. The specimen was collected in the cenote X-Nabuy (Fig 1).

## Description

The incompleteness of the single tooth found does not preclude determining its total length and height. The best-preserved portion is the crown, which is wider than long and presents a slight lateral constriction at the midpoint (Fig 7E and 7F). Both cutting edges have small serrations on the apical portion, which is discontinued in the base of the crown. The shoulders are tiny and present a few minutes of serrations. Only a little central foramen is visible on the root.

## Remarks

Hemipristid (snaggletooth sharks) upper teeth are remarkable due to the curvature and coarse serrations in the mesial and distal upper jaw cutting edges [14]. Nevertheless, the lower teeth in †*Hemipristis serra* are different: the crown is narrow, laterally sigmoid, and showing the cutting edges finely serrated and straight. Furthermore, the cutting-edge serrations are discontinued close to the base of the crown, and the shoulder is small and ornate with a few small serrations [14, 32]. Although not complete, the unique specimen here is considered as †*H. serra* due to the narrow and straight crown and discontinued serrations on the cutting edges. This species was cosmopolitan during the Miocene-Pliocene [46], and its distribution in Mexico ranged from the Middle Oligocene to the Pliocene strata [27, 28]. The unique extant species of the genus is *Hemipristis elongata* [50], which inhabits inshore and offshore on the continental and insular shelves in the Indo-West Pacific from 1 to 30 meters [14, 23].

Order Lamniformes Berg, 1937 [49]
Family †Otodontidae Glikman, 1964 [50]
Genus †*Otodus* Agassiz, 1838 [51]
†*Otodus* (*Carcharocles*) *megalodon* Agassiz, 1838 [51]
(Fig 8A–8P)

## Referred material

IGM 13926 is an incomplete tooth lacking root, and the crown shows a lateral cusplet only on the left side. IGM 13927 is an incomplete tooth, where the root and the distal cutting edge are incomplete. IGM 13928 is a complete tooth with the apex slightly turned to the left side. IGM 13929 is an almost complete tooth with a small crown height and a considerable root length. The crown is turned to the left, and the distal root lobe is lacking. The specimens IGM 13926 to IGM 13928 are from the cenote San Juan, and IGM 13929 is from the cenote Sambulá (Fig 1).

## Description

There are four teeth in distinct degrees of preservation. The total length ranges from 26.9 to 39.03 mm, and the total height ranges from 29.2 to 59.9 mm (S1 Table). The teeth are labio-lingually flattened, the crown is triangular, and the apex is not labially projected. There is a notable high degree of heterogeneity in the crown shape: in both mesial and distal views, the labial edge is straight, and the lingual edge is slightly concave except in IGM 13927 and IGM 13928, in which the lingual edge is straight and inclined. In IGM 13928 and IGM 13929, both mesial and distal cutting edges are concave. However, the mesial cutting edge in IGM 13926 is convex in the apical portion and concave at the base, while the distal edge is concave along all the

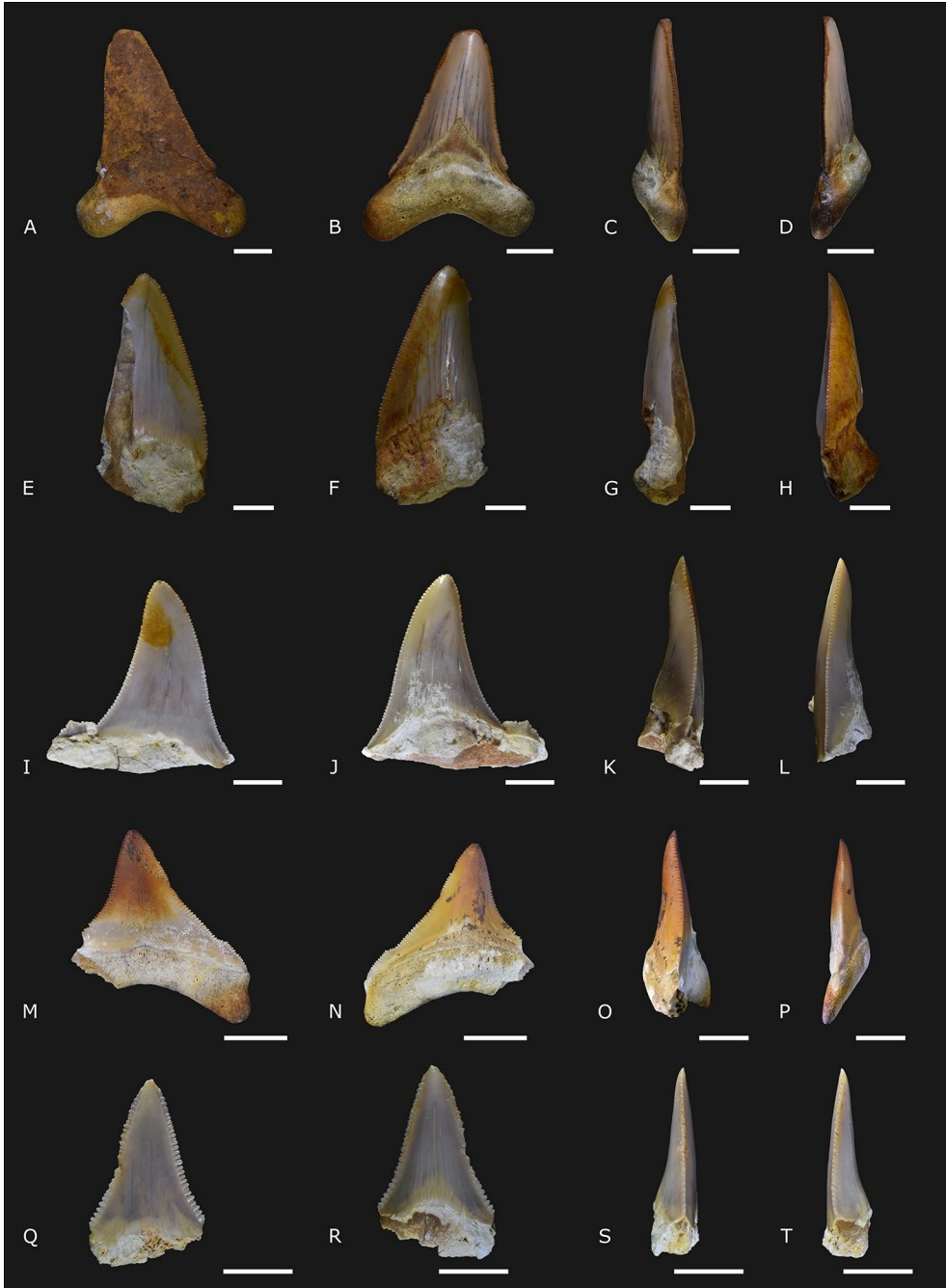

**Fig 8. †*Otodus* (*Carcharocles*) *megalodon* Agassiz, 1838 [51] specimens (A to P; IGM 13928, IGM 13927, IGM 13926, and IGM 13929) and *Carcharodon carcharias* Linnaeus, 1758 [52] (Q to T; IGM 13930) from Carrillo Puerto Formation.** (A, E, I, M, and Q) Labial view. (B, F, J, N, and R) Lingual view. (C, G, K, O, and S) Right lateral view. (D, H, L, P, and T) Left lateral view. The scales indicate 10 mm.

borders. In the specimen IGM 13927, only the mesial cutting edge is present and is convex. In IGM 13929, the crown is thinner than the others. Cusplets are present over the distal root lobe in IGM 13926 and the mesial lobe in IGM 13929. The cusplet shape is different in each specimen: in IGM 13926, the cusplet is separated from the crown by a deep notch and presents

coarse serrations with a robust medial projection, while in IGM 13929, the cusplet is almost continuous with the cutting edge, the serrations are less developed, and the medial projection is lacking. The medial portion of the ventral edge of the crown in lingual view is deep and angular, forming a sizeable triangular neck, which is best observed in IGM 13928 and IGM 13929. The root medial edge is deep and ventrally arcuate. The root lobes are robust with circular edges. The lingual protuberance of the root is pronounced. The central foramen is reduced to small circular perforations, and the transverse groove lacks.

## Remarks

The specimens are considered †*Otodus* (*Carcharocles*) *megalodon* due to the following combination of features: teeth with the crown labio-lingually flattened, cutting edges serrated and not notched, dental bands triangular and well-developed, root notched and protruding lingually at midline of the tooth, transverse groove absent in lingual root protrusion, and the basal margin of the enamel of lingual face gently arched. The small length indicates that all teeth belonged to small-sized specimens. The presence of cusplets should indicate co-occurrence of the species †*Otodus* (*Carcharocles*) *chubutensis* or †*Otodus* (*Carcharocles*) *angustidens*. However, the cusplet present in one cutting edge is a general feature found in juveniles of †*Otodus* (*Carcharocles*) *megalodon* [41, 52] that should persist in some adult forms [53].

†*Otodus* (*Carcharocles*) *megalodon* is a worldwide distributed shark during the Miocene and Pliocene, lacking occurrence only on the polar seas [41]. In Mexico, megalodon sharks are reported on Miocene and Pliocene outcrops from Baja California (Trinidad and Tirabuzón formations) and on Miocene outcrops from Palenque, Chiapas (Tulijá Formation) [27, 28, 54].

Family Lamnidae Berg,1958 [55]
Genus *Carcharodon* Smith in Müller & Henle, 1838 [56]
*Carcharodon carcharias* Linnaeus, 1758 [57]
(Fig 8Q–8T)

## Referred material

IGM 13930 is an incomplete and small tooth whose root is lacking. The specimen is from the cenote San Juan (Fig 1).

## Description

The single tooth is flattened labio-lingually, and only the crown and a small portion of the root are preserved. The crown is narrow and triangular, with the apex straight. The labial surface of the crown is almost straight in both mesial and distal views, while the lingual surface is medially concave. There is a shallow curvature of the tip of the crown to the labial portion. Both mesial and distal cutting edges have a shallow notch close to the base of the crown and are strongly serrated. The serrations are uniform along the entire surface. Cusplets are lacking. The medial curvature at the base of the labial surface is shallow. The mesial cutting edge forms an angle of about 22.7°. The root is poorly preserved. Only a tiny portion of the mesial root lobe is observed.

## Remarks

This single specimen studied is assigned to the white shark due to the absence of triangular and acute neck on the lingual surface, the coarser serrations along the cutting edges, and more labio-lingually compressed teeth. Some hypotheses indicate that †*Otodus* (*Carcharocles*) *megalodon* and *Carcharodon carcharias* are not contemporary [32]. Nevertheless, a previous co-

occurrence reported in the late Pliocene strata of Baja California [52] and the single species herein described indicates that both species occurred on both sides of the Mexican coasts during the Miocene-Pliocene.

Division Bathomorphi Cappetta, 1980 [58]

Order Myliobatiformes Compagno, 1973 [19]

Family Myliobatidae Bonaparte, 1838 [17]

Genus *Aetomylaeus* Garman, 1908 [59]

*Aetomylaeus* sp.

(Fig 9A–9L)

## Referred material

IGM 13931 is a nearly complete tooth with an eroded crown. It is completely black. IGM 13932 is a broken tooth exhibiting only about one-third of its length. The teeth are black, with the base of the root yellowish. Both specimens are from the cenote X-Nabuy (Fig 1).

## Description

The teeth are hexagonal and horizontally enlarged. The best-preserved specimen (IGM 13931) is 26.9 mm long and 4.16 mm high in occlusal view (S1 Table). Only a little remnant of the crown is preserved in all specimens. The surface of the crown is rough. The labial overhang is strongly pronounced, with the labial edge slightly concave in IGM 13931. The lingual ledge is not pronounced and is strongly enameled, with a distinct brown coloration in both specimens studied. Close to the upper surface of the polyaucorhyzus root is a horizontal series of foramina (Fig 9E and 9K). The loblets are observed as several deep grooves on the ventral view. The loblets of the lateral extremities are triangular.

## Remarks

These two teeth are classified under the genus *Aetomylaeus* due to the well-developed foramina on both labial and lingual sides and a deep polyaucorhyzus root with a weak lingual inclination [45]. The preservation of the few teeth described here does not allow us to determine the taxonomic rank at the species level. *Aetomylaeus'* fossil record includes strata from the Miocene of France [61], Miocene-Pliocene from Libya [45], and the Pungo River in North Carolina, USA [14]. The small size of these tooth plate fragments indicates that these specimens could be juveniles. Currently, the only one species known, *P. bovinus* [62], is distributed in the eastern Atlantic and the Mediterranean Sea, ranging from 10 to 100 meters from the coast [14, 63].

Family Rhinopteridae Jordan & Evermann, 1896 [64]

Genus *Rhinoptera* Cuvier, 1829 [60]

*Rhinoptera* sp.

(Fig 9M–9R)

## Referred specimen

IGM13933 is a small, incomplete tooth plate with light brown coloration and an eroded crown. The specimen is from the cenote Sambulá (Fig 1).

## Description

A single specimen with a hexagonal shape and horizontally enlarged. The teeth are 12.86 mm long and 1.9 mm in height in the occlusal view (S1 Table). Like the *Aetomylaeus* specimens found, only a little remnant of the crown is preserved. The labial overhang is less pronounced

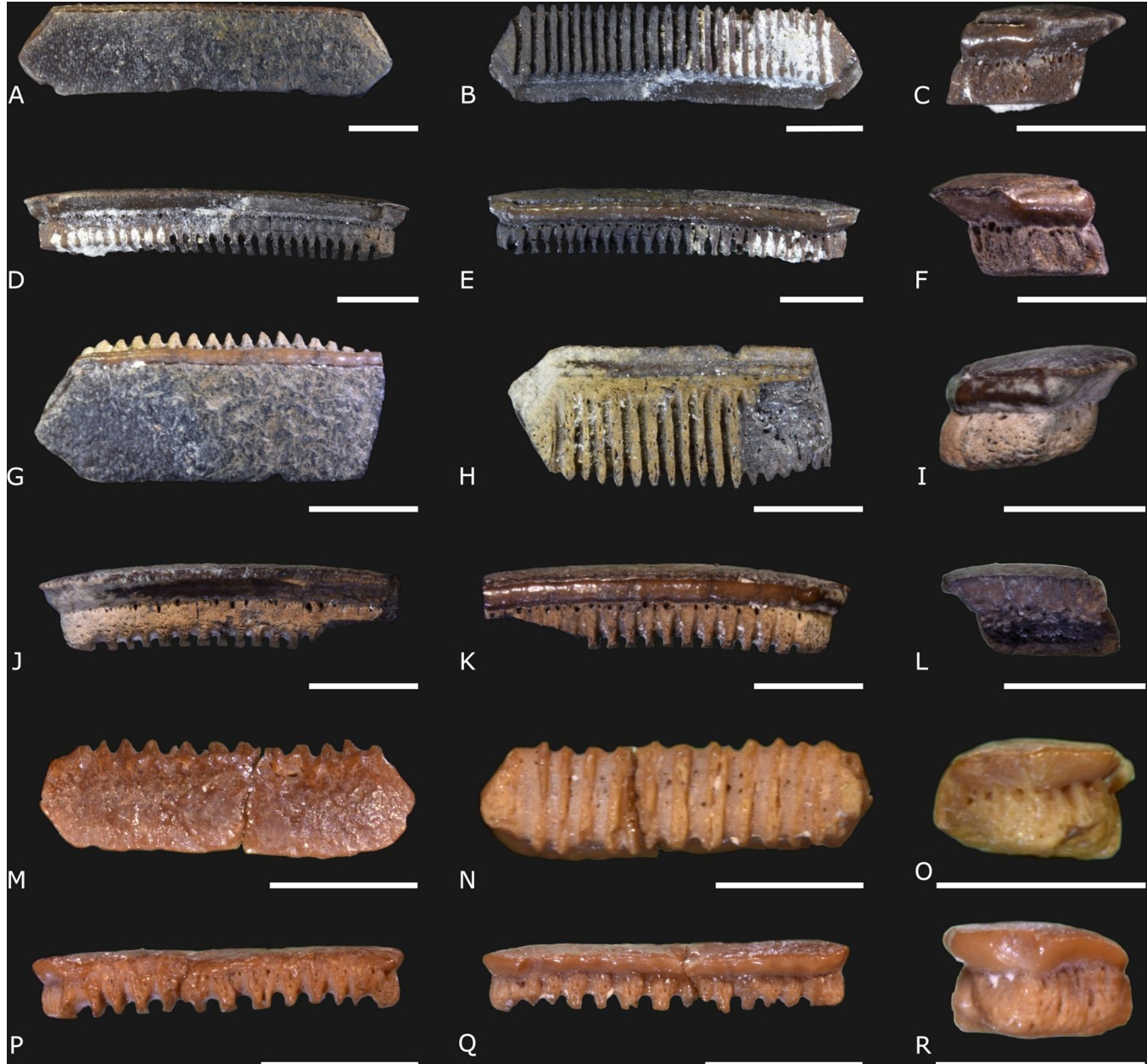

**Fig 9. *Aetomylaeus* Garman, 1908 [59] (IGM 13931 and 13932) and *Rhinoptera* Cuvier, 1829 [60] (IGM 13933) specimens from Carrillo Puerto Formation.** (A, G, and M) Occlusal view. (B, H, and N) Ventral view. (D, J, and P) Lingual view. (E, K, and Q) Labial view. (C, I, and O) Right lateral view. (F, R, and L) Left lateral view. The scales indicate 5 mm.

than the *Aetomylaeus* tooth, and the labial edge is straight. The lingual ledge is not pronounced and is strongly enameled, with the same brown coloration as the other regions of the teeth. The foramina at the dorsal surface close to the polyaucorhyzus root is vestigial and better visualized at the lateral extremities (Fig 9O and 9R). The root loblets are not deep like in the *Aetomylaeus* specimens observed and seem further away from each other (Fig 9N). The loblets of the lateral extremities of the root are triangular.

## Remarks

The single specimen herein described is considered a member of the genus *Rhinoptera* due to the absence of foramina on both labial and lingual sides of the tooth and the shallow and not uniform polyaucorhyzus with a lower number of grooves on the root. The genus *Rhinoptera* was widely distributed along marine environments around the globe during the Miocene-Pliocene. In Mexico, *Rhinoptera* species are reported from the Oligocene outcrops of the San Gregorio y El Cien Formations and Miocene strata of the San Izidro Formation, both in Baja California [27, 28].

Infraclass Teleostei Müller, 1845 [65]

Series Eupercaria Betancur-R. *et al.*, 2013 [66]

Order Tetraodontiformes Berg, 1937 [49]

Family Diodontidae Bonaparte, 1835 [67]

Genus *Chilomycterus* Brisout de Barneville, 1846 [68]

†*Chilomycterus dzonotensis* sp. nov. urn:lsid:zoobank.org:act:6DCC6451-68F8-42EE-9DAF-EB3CE798174F

(Figs 10 and 11)

## Holotype

IGM 13934 is a premaxilla with 16.5 mm in length in frontal view. The premaxilla of both sides' fuses in a unique element, in which the portion comprising the biting edge is broken (Fig 10A and 10C). The posterior bone elements, including the internal tooth plate and the posteromedial process of the premaxilla, are well-preserved. The specimen is from the cenote X-Nabuy (Fig 1).

## LSID

*Diagnosis.* †*Chilomycterus dzonotensis* sp. nov. differs from the other living and fossil burrfish specimens of the genus due to the concavity in the external lateral edges of the tooth plates that compose the trituration plate or occlusal crushing surface. This species has eight quadrangular and paired teeth, with the anterior pair being the most prominent teeth of the series. The tooth plate gradually narrows laterally until the penultimate pair of teeth. The last pair is large, like the first. Other features that distinguish †*C. dzonotensis* from the other taxa are the absence of teeth embedded into the bone matrix of the premaxilla and the lack of fusion of the upper jaw bones.

## Derivation of the name

The specific name is a latinization of the word "*d'zonot*", which the Mayan people used to refer to sinkholes. The generic and specific name means "the *Chilomycterus* species that belongs to a cenote."

## Description

The upper jaw is massive, with the premaxilla fully fused to its antimere in the midline. The anterior surface of the premaxilla has several irregular lines formed by small bony protuberances. In the fusion point of the premaxilla, there is a posteromedial process (Fig 10), which is articulated by fibrous tissues with the palatine and the ethmo-vomer in extant diodontid species (Tyler, 1980; 346) [15]. The articular premaxillary process is oval and well-developed. The premaxillary alveolar process is short and laterally directed (Fig 10).

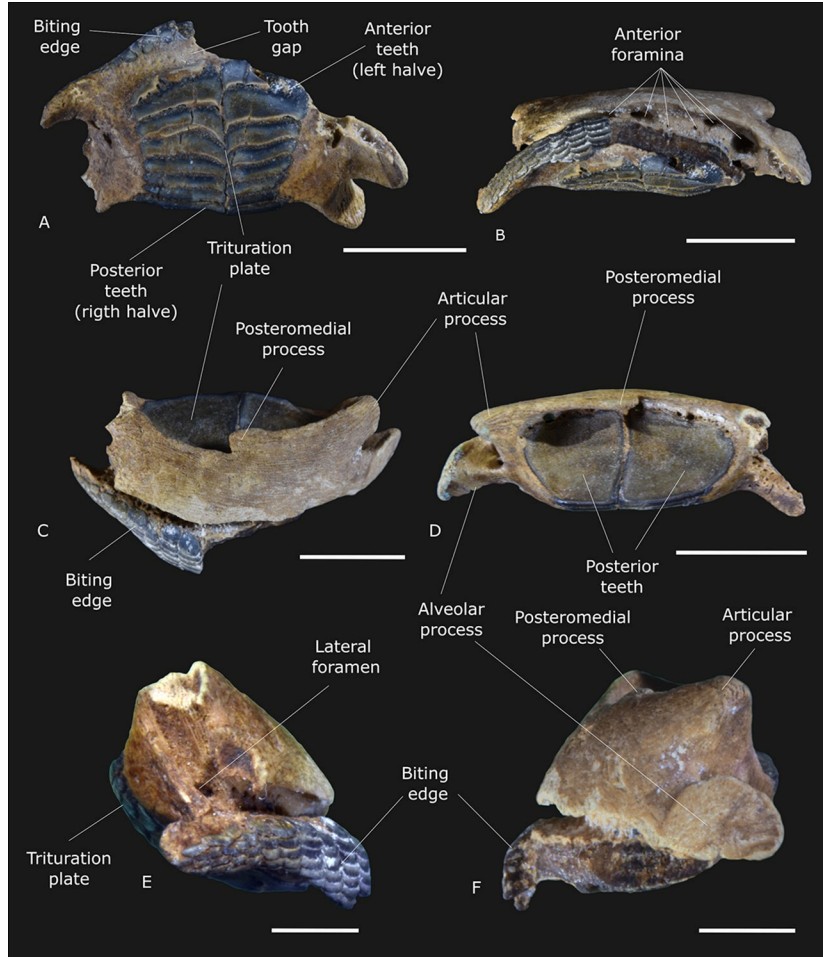

**Fig 10. Photography of the premaxilla of †*Chilomycterus dzonotensis* sp. nov. (IGM 13934).** (A) Ventral view. (B) Anterior view. (C) Dorsal view. (D) Posterior view. (E) Right lateral view. (F) Left lateral view. The scale bars indicate 10 mm.

There is a lateral foramen on each side of the premaxilla and at least seven foramina in the gap between the biting edge and the anterior surface of the premaxilla. The external foramina are well-developed and irregular (i.e., some are circular, while others are oval). The posterior foramina are small and circular (Fig 10).

The biting edge comprises several vertical series of teeth. Different from some other *Chilomycterus* species (Tyler, 1980; 365) [15], the teeth of the biting edge are not embedded in the bone matrix (Fig 10). The teeth are flat, circular in shape, and organized in vertical series. The vertical series of teeth are located laterally in the biting edge and are reduced in number, comprising about two or three teeth. The teeth of the vertical series gradually increase to the medial portion of the premaxilla, where the vertical series has seven teeth. This conformation forms a triangular and robust beak used to tear the prey [15].

The interior of the premaxilla comprises a series of eight plates accommodated in pairs (left and right halves). These plate pairs overlap. Only the posterior edges are externally visible, forming the internal trituration plate. Like the biting edge, the trituration plates in †*Chilomycterus dzonotensis* sp. nov. are different from other diodontid species since all plates are exposed while (Fig 10A and 10E), in general, they are embedded in the premaxilla (see Tyler, 1980; 365,

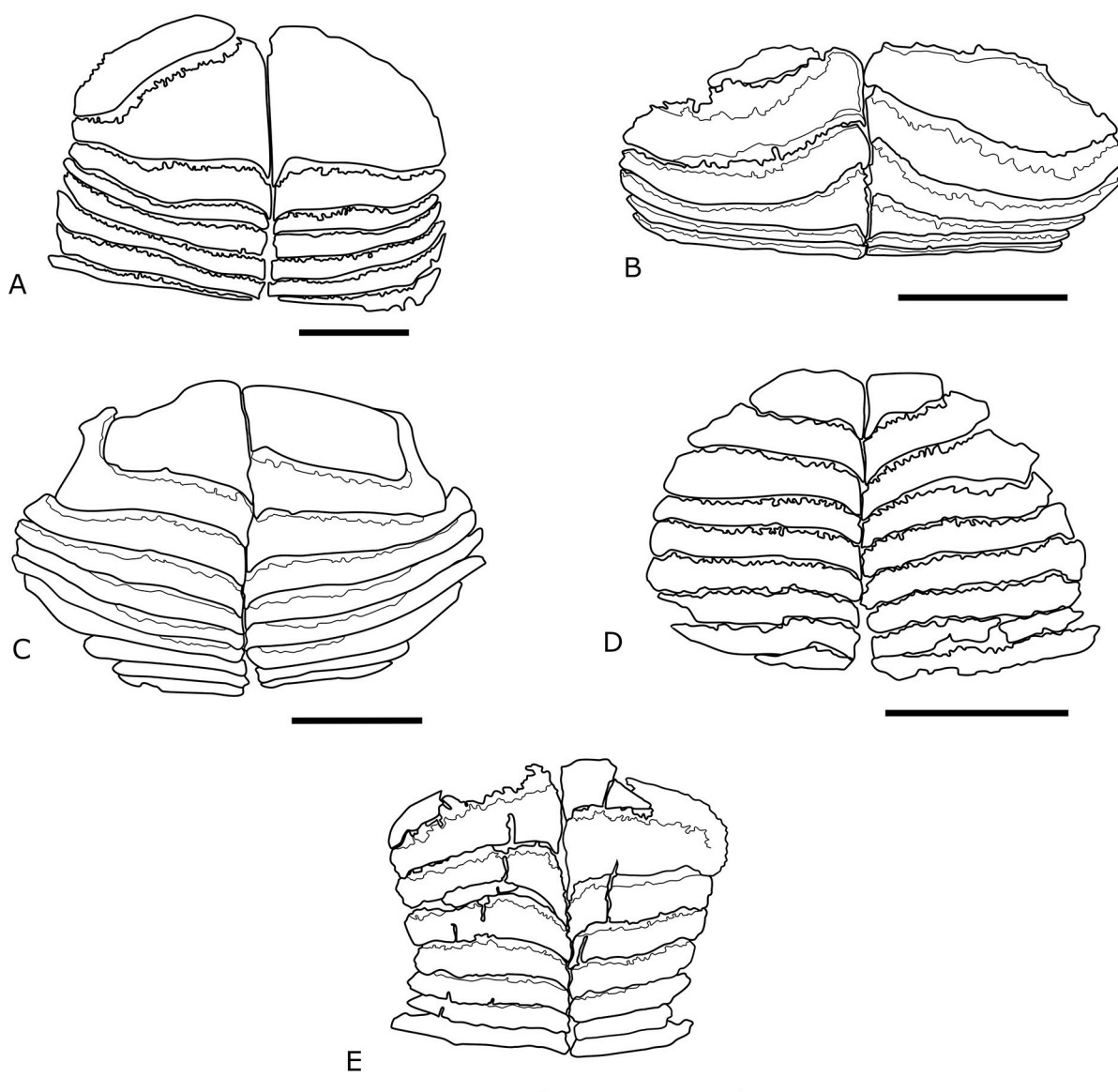

**Fig 11. Schematic drawing of the premaxilla from different Miocene diodontid species.** (A) †*Oligodiodon* sp. (modified from Schultz [81]). (B) †*Chilomycterus ferrerai* (Santos and Travassos, 1960) [82]. (C) †*C. vetus* (Leidy, 1877) [83]. (D) †*Diodon serratus* Aguilera, Carrillo-Briceño and Rodriguez, 2017 [16]. (B-D) modified from Aguilera et al. [16]). (E) †*C. dzonotensis* sp. nov (IGM 13934). The scale bars indicate 5 mm.

for example) [15]. The absence of bone rupture in the posterior region of the maxilla around the crushing plate indicates that it is a natural feature and not a result of the posterior preservation of the specimen.

In the ventral view, it is possible to observe the external surface of the trituration plate of the upper jaw. There is a gap of about 1.7 mm between the biting edge and the trituration plate (the tooth gap). From this viewpoint, the plates are almost rectangular and cover most of the upper jaw's ventral surface, reaching the premaxilla's posterior border. On the ventral surface, it is also possible to note that the plates are powerfully articulated laterally and are not parallel. The first four anterior plates are ventrally projected at the level of the internal lateral

articulation. The posterior border is rounded. The length of the last pair of teeth in the ventral view is equal to the first two, forming a concave external lateral surface of the trituration plate.

The medial surface of the teeth series that composes the trituration plate in diodontids is separated into smooth and papillae portions. Like in †*Chilomycterus ferrerai* (Aguilera et al., 2017, Fig 4) [16], †*C. dzonotensis* shows the papillae portion very pronounced. The papillae portion in the species herein described reaches almost the anterior border of the first. The smooth surface of the posterior trituration plates is more prominent than in other *Chilomycterus* species.

### Remarks

The family Diodontidae (the porcupine fish or burrfish) comprises a total of seven genera, with 18 living and 14 valid fossil (and several unidentified) species distributed in marine environments of the Atlantic, Indian, and Pacific oceans [15, 16, 69, 70]. The species †*Chilomycterus dzonotensis* sp. nov. is considered a diodontid due to the presence of the premaxilla fully fused to their opposite member, the marginal massive incisor-like teeth (the biting edge) and the occlusal crushing surface (the trituration plate) in the premaxilla, and the maxilla with a profoundly indented surface (Fig 10A–10C).

## Discussion

### The fossil diversity of the Carrillo Puerto Formation

As a result of this study, the fossil diversity from the Carrillo Puerto Formation is determined to be much greater than previously understood. The only three vertebrate species formerly cataloged and housed in a paleontological collection are a small cervid (IGM 4568) and two aquatic mammal species, †*Xenosiren yucateca* Domning, 1989 [9] and †*Corystosiren varguezi* Domning, 1990 [10], both belonging to the family Dugongidae and the subfamily Rytiodotinae [9, 10] (Table 1). The description of †*Xenosiren yucateca* is from two bone fragments (IGM 4190 and IGM 4191) found in the north portion of the cenote Kambul, about 15 km northwest of Merida, state of Yucatan (IGM-loc. 2397). While †*Corystosiren varguezi* is described from a nearly complete skull, vertebral remains, and ribs (IGM 4569) found exposed in a ground surface on Rancho Chapas, close to Tizimin municipality, northeast Yucatan (IGM-loc. 2398).

Prior work on limestones from Cenote Noc Ac and the type-locality of †*Corystosiren varguezi* mentioned the presence of other fossil organisms, including invertebrates, shark teeth, and crocodilian fragments [9, 10]. The shark species include *Carcharhinus leucas*, †*Otodus* (*Carcharocles*) *megalodon*, *Isurus hastalis*, *Hemipristis serra*, and two tiger sharks: one related to †*Galeocerdo rosaliensis* Applegate, 1978, and the other with †*G. aduncus* [10]. According to Domning [9, 10], these taxa were described by Shelton P. Applegate [10]. However, the catalog number, the current location of the specimens, and the morphological evidence that supports the taxonomic attributions are unknown (Table 1). Our study supports the presence of *C. leucas* and †*O.* (*C.*) *megalodon*, and *H. serra*. The presence of *I. hastalis* is not confirmed, and a distinct *Galeocerdo* species (†*G. mayumbensis*) is determined. Furthermore, five other shark taxa and two stingrays are reported to compose the chondrichthyan diversity of the Carrillo Puerto Formation (Table 1).

We also recognize a new teleost fish from a small upper jaw comprising the premaxilla anteriorly fused and two separated tooth plates, enough features to classify it as a member of the family Diodontidae. The crushing occlusal tooth plate shows similarities with the genus *Chilomycterus*. However, the posterior constriction of the tooth plate and the condition of all teeth exposed are not observed in other fossil or extant diodontid species. This finding represents an

**Table 1. Vertebrate fossil record reported or mentioned from the Carrillo Puerto Formation.**

| Taxon | Catalog number | Reference |
|---|---|---|
| *Carcharhinus brachyurus* | IGM 13913, IGM 13914, IGM 13915, IGM 13916 | This study |
| *Carcharhinus leucas* | IGM 13917, IGM 13918, IGM 13919, IGM 13920 | Domning (1990) [10], this study |
| *Carcharhinus macloti* | IGM 13921 | This study |
| *Carcharhinus perezii* | IGM 13922 | This study |
| *Carcharhinus* sp. | IGM 13923 | This study |
| *Galeocerdo* sp. | - | Domning (1989) [9] |
| *Galeocerdo* aff. *rosaliensis* | - | Domning (1990) [10] |
| *Galeocerdo mayumbensis* | IGM 13924 | This study |
| *Hemipristis serra* | IGM 13925 | Domning (1989) [9], this study |
| *Isurus hastalis* | - | Domning (1989, 1990) [9, 10] |
| *Otodus* (*Carcharocles*) *megalodon* | IGM 13926, IGM 13927, IGM 13928, IGM 13929 | Domning (1989, 1990) [9, 10], this study |
| *Carcharodon carcharias* | IGM 13930 | This study |
| *Aetomylaeus* sp. | IGM 13931, IGM 13932 | This study |
| *Rhinoptera* sp. | IGM 13933 | This study |
| *Chilomycterus dzonotensis* sp. nov. | IGM 13934 | This study |
| *Corystosiren varguezi* | IGM 4569 | Domning (1990) [10] |
| *Xenosiren yucateca* | IGM 4190 | Domning (1989) [9] |
| Crocodilia undetermined | - | Domning (1990) [10] |
| Cervidae undetermined | IGM 4568 | Domning (1989) [9] |

increment in the diversity and distribution of the family Diodontidae during the Miocene-Pliocene in the Western Atlantic Sea domains.

## The new burrfish fossil species

Jaw elements are the most common bone structures preserved in diodontid fossils. The family Diodontidae has at least 62 ancient records [71], with the oldest report belonging to the Late Cretaceous strata from Brazil, precisely in the Maastrichtian strata from Gramame Formation, Paraiba state [72]. The species herein described is easily differentiated from the Mesozoic Brazilian diodontid specimen due to the reduced number (8) of internal plates in the premaxilla (see Gallo et al. [72], Fig 2).

The diodontid diversification is notable during the Paleogene, with several genera described, such as †*Heptadiodon* Bronn, 1855 [73], †*Eodiodon* Casier, 1952 [74], †*Progymnodon* Dames, 1883 [75], †*Prodiodon* Le Danois, 1955 [76], †*Zignodiodon* Tyler and Santini, 2002 [77], †*Pshekhadiodon* Bannikov and Tyler, 1997 [78], and †*Oligodiodon* Sauvage 1873 [79]. Except for †*Heptadiodon* and †*Zignodon fornasieroae* Tyler and Santini, 2002 [77], in which the information provided about the upper jaw is scarce to compare with †*C. dzonotensis*, the other Paleogene taxa, can be distinguished from the species described here.

The genus †*Eodiodon* is the unique diodontid taxon where the tooth plate is lacking ([75], Fig 1A). The genus †*Progymnodon* has tooth plates almost in contact with the biting edge, and the tooth plates are semicircular ([74], Fig 1B); ([80], Plate I, Fig 1). The species †*Psekhadiodon parini* Bannikov and Tyler 1997 [78] have the first trituration plates large, as found in the species described. However, there are only four pairs of tooth plates in the premaxilla, and the lateral edges of the plates are convex. Inside the genus †*Prodiodon*, only †*P. tenuispinus* has the

internal tooth plate of the lower jaw preserved. These plates are parallel and gradually increase in length ([77], Fig 28), contrasting with †*C. dzonotensis*, which has the plates unparallel with a concave lateral edge.

During the Neogene, the number of diodontid genera was reduced. However, it was the moment when species of extant genera rose. Inside the Cenozoic groups, the genus †*Oligodiodon* is unique to pass through the Paleogene-Neogene boundary [81]. Like †*C. dzonotensis* sp. nov., †*Oligodiodon* have at least seven pairs or trituration plates (Fig 11). However, the teeth are strongly imbricated, and the width of the plates is equal along all the occlusal surfaces (Fig 11A). In †*Chilomycterus dzonotensis* sp. nov., the trituration plates are not strongly imbricated, and the plate width is unequal, with the external lateral surface forming a constriction in the occlusal surface (Fig 11E).

The differentiation of fossil species of extant genera is based mainly on the position of the trituration internal plates about the biting edge. According to Aguilera et al. [16], the crushing plates of *Diodon* are on the posterior portion of the occlusal surface, while in *Chilomycterus*, the internal plates are close to the biting edge. Adult species are also different in the number of internal plates. Large *Diodon* species have about 35 trituration plates, while adult *Chilomycterus* has only 18 plates [15, 16].

The length of these structures in the species herein studied indicate a small-sized specimen with about 10 cm ([15], Fig 302; [16], S1 Fig). Therefore, taxonomic assignment is not possible through the number of trituration plates. Nevertheless, the approximation of the trituration tooth plate with the biting edge (see Fig 10 and [16], Fig 4), the alternate position of the tooth plates, and the teeth conformation in cross-section (see Fig 11 and [15], Figs 294 to 300) suggest that the species herein described belongs to the genus *Chilomycterus*.

*Chilomycterus* is the most diverse in the diodontid fossil record, especially in the Proto-Caribbean region during the Miocene [16]. The American *Chilomycterus* fossil record includes †*C. circunflexus* Leriche, 1942 [84] from La Habana, Cuba, and form Florida, USA, †*C. exspectatus* Aguilera, Carrillo-Briceño, and Rodriguez, 2017 [16], †*C. tyleri* Aguilera, Carrillo-Briceño, and Rodriguez, 2017 [16, 85], †*C. gatunensis* Toula, 1909 [85], from Gatun Formation, San Judas Tadeo, Colon, Panama, †*C. kugleri* Casier, 1958 [86] from Gross Morne Formation, and †*C. vetus* Leidy, 1877 [83] from Tamana Formation, both from Trinidad, †*C. ferrerai* Santos & Travassos, 1960 [82], from several localities of the Pirabas Formation, at Para State, Northern Brazil, and Canture Formation, Venezuela. One unidentified *Chilomycterus* species from the Tuira Formation, Panama, and other unidentified species from the Guajira Peninsula, Colombia [16]. No living or fossil *Chilomycterus* species have a constriction on the external edge of the occlusal crushing surface ([16]; Figs 4–8), a diagnostic feature of *Chilomycterus* specimen here studied.

Although diodontids were very diverse during the Neogene in the tropical region comprising the northern portion of South America, Central, and North America ([16], Fig 9), †*C. dzonotensis* is the first diodontid fossil from the Miocene-Pliocene strata of the Gulf of Mexico.

## Fish diversity significance

Previous studies on the lithological features and fossil invertebrates indicated a shallow shelf depositional environment with internal neritic conditions and middle neritic influence on the Carrillo Puerto Formation [7]. Comparing the distribution of the Carrillo Puerto Formation with the Eocene Chichén Itzá Formation (Fig 1) and the current littoral composition of the Yucatan Peninsula in present-day is evident a marine regression during the Cenozoic this region (Fig 1). The marine regression provided extensive periods of warm shallow marine water associated with a coral reef system, an excellent refuge for tiny organisms, such as teleost

specimens, sharks, and stingrays. Our results are in complete accordance with the paleoenvironmental interpretation for the Carrillo Puerto Formation since all taxa are marine species that usually inhabit warm shallow waters not exceeding 350 meters in depth and include reef-associated species, such as *Carcharhinus brachyurus* and *C. perezii.*

The teleost fish and the megalodon shark indicate the presence of small specimens in the Carrillo Puerto Formation. The upper jaw of †*Chilomycterus dzonotensis* sp. nov., for example, has only 16.5 mm in horizontal length, and the shape and number of the tooth plates in the occlusal surface indicate that the total length of the specimen should be about 10 cm, according to Tyler [15]. The maximum length of extant *Chilomycterus* species ranges from 27.9 to 69.7 cm [40]. The biggest megalodon tooth in the Carrillo Puerto Formation is about 39 mm in total length, and two specimens show a lateral cusplet only on one side.The presence of lateral cusplets was initially proposed to separate the species †*Otodus* (*Carcharocles*) *chubutensis* and †*O.* (*C.*) *angustidens* from †*O.* (*C.*) *megalodon*. Nevertheless, recent studies using life-stage tooth morphology and total length estimates of several megalodon specimens indicate this is a standard feature found in juvenile specimens of megalodon sharks [41].

Another point to consider with the fish fauna from Carrillo Puerto Formation is the repeated co-occurrence of the white shark and megalodon on Mexican coasts during the Miocene-Pliocene. Traditionally, it is believed that *Charcharodon carcharias* and †*Otodus* (*Carcharocles*) *megalodon* have had allopatric distributions [32]. However, in Mexico, this co-occurrence is reported from the Pliocene sediments of Baja California [28, 52]. Our results reinforce the co-occurrence of white and megalodon sharks in America during the Miocene-Pliocene.

Like megalodon sharks, the extinction of other taxa in American marine environments after the Miocene-Pliocene is observed in the genus *Aetomylaeus* and the shark species *Carcharhinus macloti*. Different from megalodon, which was extinct after the Pliocene, extant organisms of *Aetomylaeus* and *C. macloti* are currently distributed in marine shallow habitats of tropical and temperate regions, but not on the North Atlantic coasts of America [39]. The fossil record on the Carrillo Puerto Formation reinforces the presence of these species on the Western Atlantic coast during the Miocene-Pliocene. It is a strong indicator of the loss of habitat of some species from local extinction. A more detailed study is necessary to understand the nature of these extinctions after the Miocene-Pliocene. A putative hypothesis to explain it could be that the repetitive glaciation events that occurred then [87] were responsible for the extinction and reduction of distribution of some taxa on Atlantic coasts during the Late Cenozoic.

## Conclusion

The underwater strata from the Carrillo Puerto Formation inside the sinkholes are remarkable evidence of the gradual marine regression that occurred during the Late Cenozoic that was responsible for the formation of the current Yucatan Peninsula. Here, we show the Carrillo Puerto Formation fish biodiversity, which is composed of 1) extinct taxa, such as the megalodon shark and the new burrfish species †*Chilomycterus dzonotensis* sp. nov. 2) extant organisms that are currently distributed in marine Mexican coasts, such as the white shark, and 3) extant species that are not currently distributed on western Atlantic domains, such as the hardnose shark and the bull ray.

The fish fauna discovered represents an essential increment in vertebrate diversity during the Late Miocene and Early Pliocene of the Gulf of Mexico since several described taxa had yet to be previously reported. Although some shark species were previously mentioned in the Carrillo Puerto Formation [9, 10], the specimens used for the identification are lost and not

cataloged. Therefore, the organisms shown here are unique physical examples formally deposited in a public paleontological collection for future investigations.

The fossil diversity supports the shallow marine paleoenvironment associated with a coral reef system for the Carrillo Puerto Formation. Furthermore, the small size of some specimens studied, together with the presence of cusplets in megalodon shark teeth, indicates the possible presence of small specimens of elasmobranchs and teleost on the depositional sediments that compose the Carrillo Puerto Formation today. Finally, species currently not distributed in the Gulf of Mexico point out local extinctions during the Late Cenozoic period on the Northwestern Atlantic, probably associated with glaciation events.

## Supporting information

**S1 File. Video showing the side-mount diving technique used during the dive inside the Cenote San Juan.** Underwater recording made with a GoPro Hero 9 black and Bigblue LED video lights.
(MP4)

**S2 File. Video showing the sample technique using a chisel and hammer inside the Cenote San Juan.** Underwater recording made with a GoPro Hero 9 black and Bigblue LED video lights.
(MP4)

**S1 Table. Measures and proportions related to the total height in the elasmobranch specimens studied.** The asterisk symbol (*) indicates uncertainty, and the interrogation symbol (?) indicates that the measurement was impossible to determine due to specimen preservation. The symbol N/A means that this feature is not applicable in this specimen.
(DOCX)

## Acknowledgments

We are grateful to the authorities responsible for the Cenotes and the *Secretaría de Desarrollo Sustentable* (SDS) of Yucatan and the *Subdirección de Arqueología Subacuática* (SAS) of the *Instituto Nacional de Antropología e Historia* (INAH) for providing permits to collect material. Thanks to Pablo Bayardo Gómez (Overhead Diving Mexico) and Michael Silva Netto (CCRXTREME) for cave dive support. Thanks to Myriam Miranda and Lourdes Fernández for the material transport. Thanks to Jesús Alvarado Ortega and Violeta Amparo Romero Mayén for providing information about the previously described species from the Carrillo Puerto Formation and for cataloging and accommodating the specimens studied in the National Collection of Paleontology of Mexico, UNAM. Thanks to the anonymous reviewers for the improvement in the final version of the manuscript. The work is part of the Underwater Archaeological Atlas project, coordinated by Helena Barba Meinecke (helenabarbamei@yahoo.com.mx) from *Subdirección de Arqueología Subacuática* (SAS) of the *Instituto Nacional de Antropología e Historia* (INAH).

## Author Contributions

**Conceptualization:** Kleyton M. Cantalice, Hugo E. Salgado-Garrido.

**Data curation:** Kleyton M. Cantalice, Hugo E. Salgado-Garrido, Erick Sosa-Rodríguez, Kay Vilchis-Zapata, Gerardo González-Barba.

**Formal analysis:** Kleyton M. Cantalice, Gerardo González-Barba.

**Funding acquisition:** Kleyton M. Cantalice.

**Investigation:** Kleyton M. Cantalice, Hugo E. Salgado-Garrido, Gerardo González-Barba.

**Methodology:** Kleyton M. Cantalice, Erick Sosa-Rodríguez, Kay Vilchis-Zapata, Gerardo González-Barba.

**Resources:** Hugo E. Salgado-Garrido, Erick Sosa-Rodríguez, Kay Vilchis-Zapata.

**Validation:** Hugo E. Salgado-Garrido, Erick Sosa-Rodríguez, Kay Vilchis-Zapata, Gerardo González-Barba.

**Visualization:** Erick Sosa-Rodríguez, Kay Vilchis-Zapata.

**Writing – original draft:** Kleyton M. Cantalice, Hugo E. Salgado-Garrido, Gerardo González-Barba.

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
