## [Decision Letter · Decision Letter 0]

24 Jun 2024

PONE-D-24-07932Underwater paleontology inside cenotes reveals the Miocene-Pliocene fish diversity in the Yucatan Peninsula, southeast MexicoPLOS ONE

Dear Dr. Cantalice,

Thank you for submitting your manuscript to PLOS ONE. After careful consideration, we feel that it has merit but does not fully meet PLOS ONE’s publication criteria as it currently stands. Therefore, we invite you to submit a revised version of the manuscript that comprehensively addresses the points raised during the review process.

We look forward to receiving your revised manuscript.

Kind regards,

Michael Schubert

Academic Editor

PLOS ONE

2. Manuscripts reporting paleontology and archaeology research must adhere to our policies described at http://journals.plos.org/plosone/s/submission-guidelines#loc-paleontology-and-archaeology-research. Specifically, appropriate identification numbers for the human remains, specimens and/or samples should be provided, and the data used in the study should be publicly deposited or made accessible for replication of the study. If applicable, please ensure permission to conduct destructive sampling was obtained. Only proceed to acceptance after these requirements are met.

In keeping with PLOS ONE’s standard policies, please check that you have not published with any of the authors of this submission within the last 5 years. The journal asks Academic Editors to recuse themselves from handling a manuscript if they have a potential competing interest, including recent co-publications. For more information on PLOS ONE’s competing interests policy for Editors see here: https://journals.plos.org/plosone/s/competing-interests#loc-who-must-declare-competing-interests.

If you have any concerns, please contact us at plosone@plos.org.

“KMC. Grant Number: IA206123. Dirección Genral de Asuntos del Personal Académico (DGAPA) - Programa de Apoyo a Proyectos de Investigación e Innovación Tecnológica (PAPIIT).

5. One of the noted authors is a group or consortium [Underwater Archaeological Atlas project^]. In addition to naming the author group, please list the individual authors and affiliations within this group in the acknowledgments section of your manuscript. Please also indicate clearly a lead author for this group along with a contact email address.

6. We note that Figure 1 in your submission contain [map/satellite] images which may be copyrighted. All PLOS content is published under the Creative Commons Attribution License (CC BY 4.0), which means that the manuscript, images, and Supporting Information files will be freely available online, and any third party is permitted to access, download, copy, distribute, and use these materials in any way, even commercially, with proper attribution. For these reasons, we cannot publish previously copyrighted maps or satellite images created using proprietary data, such as Google software (Google Maps, Street View, and Earth). For more information, see our copyright guidelines: http://journals.plos.org/plosone/s/licenses-and-copyright.

Reviewers' comments:

Reviewer's Responses to Questions

**Comments to the Author**

1. Is the manuscript technically sound, and do the data support the conclusions?

Reviewer #1: Yes

Reviewer #2: Yes

2. Has the statistical analysis been performed appropriately and rigorously? 

Reviewer #1: N/A

Reviewer #2: N/A

3. Have the authors made all data underlying the findings in their manuscript fully available?

Reviewer #1: Yes

Reviewer #2: No

4. Is the manuscript presented in an intelligible fashion and written in standard English?

Reviewer #1: Yes

Reviewer #2: No

5. Review Comments to the Author

Reviewer #1: This manuscript presents a species inventory for a new Mio-Pliocene locality in the Yucatan Peninsula, expanding our understanding of the shark and teleost diversity. Of particular interest is that the locality is only accessible diving.

Results indicate that the vertebrate fauna recovered from this locality is consistent with invertebrate faunas, which predict an shallow shelf neritic zone of deposition. This work further expands the taxonomic diversity of the formation. It also points out that these specimens recovered are disproportionately smaller than from other Mio-Pliocene localities. This lead authors to hypothesize that the locality was a shelter for sub-adult individuals (aka, a nursery).

As a reader, I'd like to know more about the evidence supporting the sub-adult nursery hypothesis. It is among the most interesting aspects of the paper and should come earlier in the discussion. The species inventory and table 1 can be used to support that all specimens and taxa are in fact small. Then go into why all of these might be small due to ontogeny vs some other size-variability considerations. You currently have one example for Otodus megeladon, but what else?

In addition, the second most interesting thing is the new taxon of burrfish. This warrants more time in the discussion. To some extent, I'd consider moving some of what is listed in remarks (pages 34-37) to the actual discussion section for the entire paper, to get the most readership.

Other structural considerations:

The use of Table 1 sooner may help you organize the Systematic Paleontology descriptions. Then you can reduce the descriptions to only the new ones from this study - while validating why it isn't something that was previously published by Domning (1989/1990).

Taxonomy - I'm most familiar with batoids and teleosts, thus a bit of a heavy discussion on specimens in figure 8.

Pteromylaeus is a synonym of Aetomylaeus (Garman 1908). The description and morphology of that specimen is consistent with it's placement in Aetomylaeus as well. Incredibly fine tooth roots, as narrow as the space between roots, step like displacement of roots in the labial direction. No interlocking ridge and grove as in rhinobatos.

For the sharks teeth, I will need to defer to another reviewer to confirm synonymy and correct placement.

Discussion - The discussion is a little underwhelming. There is too much review of what was previously known and the significance of the fish biodiversity comes late. See notes above. I want to read more about the impact of these fossils and about the locality.

Figures - At minimum for Figure 9 of the new taxon, you need to add labels and point out any apomorphic trait. Ideally, you will do the same for the other teeth as well.

Specific Line Comments:

page 3, line 74. Change "evidence the presence" to "preserve"

page 30, line 643. Indicate what anterior and posterior.

page 31, line 659, add the word burrfish before the word specimens.

page 31, lines 658-661. This is a long sentence that can be broken up for clarity.

page 31, line 662. "last pair" - is this anterior or posterior?

page 32, line 673, change "opposite" to "antemere"

page 32, line 687, change "teeth' vertical series" to "vertical series of teeth"

page 32, line 690, add a comma after beak

page 37, line 791. Porrly known because... Add more information.

page 38, table 1: be consistent in the placement of the term "this study" so that it is always first before another reference

page 39, line 805: Change first word, "On" to "Prior work on"

page 39, line 817. change "presents" to "comprises"

page 39, line 821. The word "exposed" - this comes up in a few places and needs a better definition and label associated with illustrations/figures. Describe wgat that actually means and how you know that it is real morphology vs taphonomy.

page 41, line 853, delete the word 'interesting'

page 41, line 861. What does "posterior extinction" mean?

page 42, lines 878-883. Awkward sentence structure.

Reviewer #2: The authors describe the fish fossil assemblage recovered from the Miocene-Pliocene deposits of the Carrillo Puerto Formation located in the Yucatan Peninsula, Mexico. The analyzed specimens include shark taxa from the Mexican coast and a newly identified diodontid fish. The assemblage, along with the environmental interpretation of the deposits, suggests that these taxa inhabited a shallow water habitat, possibly serving as a nursery area.

I thoroughly enjoyed this paper and think this manuscript deserves to be published on Plos One, particularly for the sampling method. It blew my mind when I realized that you scuba-dived in caves to recover the specimens — an incredibly challenging and impressive feat. I here recommend this manuscript for publication in Plos One after minor revisions given the following general comments:

1) Use standard terminology to describe shark teeth and guide readers through each morphological and morphometric feature. Please include a schematic figure in the methods section to explain these traits. Additionally, the proportions you reported do not sum to 100%, which raises concerns about your measurement methods. Lastly, I have never seen a shark tooth dimension described as “depth.” Do you mean height? Please reword this.

2) You cannot definitively state that the Carrillo Puerto Formation was a nursery area without quantitative analysis to support this hypothesis. Include all measurements in the supplementary materials and reference this table to substantiate your interpretation. If you aim to corroborate this hypothesis, you must include body size estimates and the range of body sizes for modern analogs at different ontogenetic stages for comparison. Otherwise, stay general.

3) The main text needs polishing. Some words are misspelled, while others seem out of context. I have included suggestions to reword some paragraphs, but I recommend carefully reviewing the manuscript before the resubmission.

These are general comments, with more specific ones provided in the main text. Good luck with the revision and I look forward to seeing your manuscript published on Plos One.

6. PLOS authors have the option to publish the peer review history of their article (what does this mean?). If published, this will include your full peer review and any attached files.

Reviewer #1: No

Reviewer #2: No

---

## [Author Response · Author response to Decision Letter 0]

7 Sep 2024

5. Review Comments to the Author

Reviewer #1: This manuscript presents a species inventory for a new Mio-Pliocene locality in the Yucatan Peninsula, expanding our understanding of the shark and teleost diversity. Of particular interest is that the locality is only accessible diving.

Results indicate that the vertebrate fauna recovered from this locality is consistent with invertebrate faunas, which predict an shallow shelf neritic zone of deposition. This work further expands the taxonomic diversity of the formation. It also points out that these specimens recovered are disproportionately smaller than from other Mio-Pliocene localities. This lead authors to hypothesize that the locality was a shelter for sub-adult individuals (aka, a nursery).

As a reader, I'd like to know more about the evidence supporting the sub-adult nursery hypothesis. It is among the most interesting aspects of the paper and should come earlier in the discussion. The species inventory and table 1 can be used to support that all specimens and taxa are in fact small. Then go into why all of these might be small due to ontogeny vs some other size-variability considerations. You currently have one example for Otodus megeladon, but what else?

Response. We include the nursery hypothesis later in the text because we decided to consider taxonomic aspects first. The megalodon teeth are a clear example of a sub-adult species because of the secondary cusplets. Nevertheless, the small size is a feature of all teeth specimens in the Carrillo Puerto Formation. 

In addition, the second most interesting thing is the new taxon of burrfish. This warrants more time in the discussion. To some extent, I'd consider moving some of what is listed in remarks (pages 34-37) to the actual discussion section for the entire paper, to get the most readership.

Response. Suggestion accepted.

Other structural considerations:

The use of Table 1 sooner may help you organize the Systematic Paleontology descriptions. Then you can reduce the descriptions to only the new ones from this study - while validating why it isn't something that was previously published by Domning (1989/1990).

Response. Domning (1989, 1990) just mentioned the possible presence of some species we described here. Although previously mentioned, this is the first time fish species from Carrillo Puerto have been formally reported, including catalog number, diagnosis features, etc. Therefore, we believe that all descriptions must remain in the manuscript. 

Taxonomy - I'm most familiar with batoids and teleosts, thus a bit of a heavy discussion on specimens in figure 8.

Pteromylaeus is a synonym of Aetomylaeus (Garman 1908). The description and morphology of that specimen is consistent with it's placement in Aetomylaeus as well. Incredibly fine tooth roots, as narrow as the space between roots, step like displacement of roots in the labial direction. No interlocking ridge and grove as in rhinobatos. For the sharks teeth, I will need to defer to another reviewer to confirm synonymy and correct placement.

Response. Changed. Pteromylaeus to Aetomylaeus. Thank you.

Discussion - The discussion is a little underwhelming. There is too much review of what was previously known and the significance of the fish biodiversity comes late. See notes above. I want to read more about the impact of these fossils and about the locality.

Response. We believe the discussion is the moment to compare our results with the previously published works. Because of this, we put the information about biodiversity first and the contribution of our study afterward. For example, the first paragraph in the fossil diversity section of the Carrillo Puerto Formation talks about the formally described species. The second concerns the not formally described ones and our contribution to understanding the Carrillo Puerto fossil elasmobranch diversity. The last paragraph is a small discussion about the new diodontid species described. 

 After this section, the discussion continues with a detailed discussion of the diodontid family (such as suggested by the reviewer) and an explanation of the diagnostic features that allow us to determine this fish as a new diodontid species belonging to the genus Chilomycterus. The information about the impact of these fossils and the locality was in the Fish diversity significance section of the Discussion. Our study aims to provide the first formal description of the fossil fish biodiversity from the Carrillo Puerto Formation. This study is essential because this first fossil fish inventory supports determining that the fish diversity is more significant than previously thought. The small-sized teeth observed (together with sub-adult features found in megalodon teeth) suggest that this locality probably was a nursery. Unfortunately, the small number of well-preserved specimens does not allow us to perform a more detailed statistical study to give support for our hypothesis about the past environment in the Carillo Puerto Formation. We hope to increase the number of specimens posteriorly.

Figures - At minimum for Figure 9 of the new taxon, you need to add labels and point out any apomorphic trait. Ideally, you will do the same for the other teeth as well.

Response. The labels were added. 

Specific Line Comments:

page 3, line 74. Change "evidence the presence" to "preserve"

page 30, line 643. Indicate what anterior and posterior.

page 31, line 659, add the word burrfish before the word specimens.

page 31, lines 658-661. This is a long sentence that can be broken up for clarity.

page 31, line 662. "last pair" - is this anterior or posterior?

page 32, line 673, change "opposite" to "antemere"

page 32, line 687, change "teeth' vertical series" to "vertical series of teeth"

page 32, line 690, add a comma after beak

page 37, line 791. Porrly known because... Add more information.

Response. The detailed information about the Carrillo Puerto follows this sentence

page 38, table 1: be consistent in the placement of the term "this study" so that it is always first before another reference

page 39, line 805: Change first word, "On" to "Prior work on"

page 39, line 817. change "presents" to "comprises"

page 39, line 821. The word "exposed" - this comes up in a few places and needs a better definition and label associated with illustrations/figures. Describe wgat that actually means and how you know that it is real morphology vs taphonomy.

Response. An explanation of our conclusion that this feature is a natural morphological feature and not taphonomy was included in the description section. At the end of the fourth paragraph. 

page 41, line 853, delete the word 'interesting'

page 41, line 861. What does "posterior extinction" mean?

Response. We are trying to explain that the megalodon was extinguished posteriorly. The sentence was removed. 

page 42, lines 878-883. Awkward sentence structure.

Response. The sentence was changed.

Reviewer #2: The authors describe the fish fossil assemblage recovered from the Miocene-Pliocene deposits of the Carrillo Puerto Formation located in the Yucatan Peninsula, Mexico. The analyzed specimens include shark taxa from the Mexican coast and a newly identified diodontid fish. The assemblage, along with the environmental interpretation of the deposits, suggests that these taxa inhabited a shallow water habitat, possibly serving as a nursery area.

I thoroughly enjoyed this paper and think this manuscript deserves to be published on Plos One, particularly for the sampling method. It blew my mind when I realized that you scuba-dived in caves to recover the specimens — an incredibly challenging and impressive feat. I here recommend this manuscript for publication in Plos One after minor revisions given the following general comments:

1) Use standard terminology to describe shark teeth and guide readers through each morphological and morphometric feature. Please include a schematic figure in the methods section to explain these traits. Additionally, the proportions you reported do not sum to 100%, which raises concerns about your measurement methods. Lastly, I have never seen a shark tooth dimension described as “depth.” Do you mean height? Please reword this.

Response. The terminology was standardized. The schematic figure was included. The proportions are more than 100% because of the nature of shark teeth. The schematic figure explains why these proportions exist. All the manuscript was reworded with the word height. Thank you.

2) You cannot definitively state that the Carrillo Puerto Formation was a nursery area without quantitative analysis to support this hypothesis. Include all measurements in the supplementary materials and reference this table to substantiate your interpretation. If you aim to corroborate this hypothesis, you must include body size estimates and the range of body sizes for modern analogs at different ontogenetic stages for comparison. Otherwise, stay general.

Response. The table was performed and included as supplementary data. 

3) The main text needs polishing. Some words are misspelled, while others seem out of context. I have included suggestions to reword some paragraphs, but I recommend carefully reviewing the manuscript before the resubmission.

Response. The manuscript was revised, and the text was polished.

These are general comments, with more specific ones provided in the main text. Good luck with the revision and I look forward to seeing your manuscript published on Plos One.

Main text responses: 

Lines 33-35. Changed to “hosts a unique underwater karstic system made of galleries connected by multiple sinkholes, locally called cenotes.”

Lines 37-38. Changed to "of these deposits and expose the fossil assemblage."

Line 42. Changed to “extinct taxa.” 

Lines 44-45. Changed to “taxa that are not currently distributed in Mexican coasts, such as Carcharhinus macloti and representatives of the genus Pteromylaeus.”

Line 57. Removed the word “solutions” and changed “sinkhole” to “sinkholes”.

Line 59. Added the word “Mexico”.

Lines 63-67. Sentence removed.

Lines 68-69. Sentence removed. 

Lines 69-70. Removed “Earth science research indicates that”

Line 73. Changed “A great portion” to “Most”

Line 73. Removed “is formed by coquina strata, which”.

Lines 79-80. Moved to line 59.

Line 79. Changed “prospections” to “sampling”.

Lines 83-88. Changed to “The Carrillo Puerto Formation outcrops in the Yucatan Peninsula over an area of about 8800 Km2, which includes the central and eastern part of the state of Quintana Roo, the central region of Yucatán, and northeastern Campeche. It comprises limestones that classified into…”

Line 87. Changed “five” to “5”.

Line 96. Changed “masl” to “m.a.s.l.”.

Line 104. Removed the word “impure”.

Line 114. Changed “Formations” to “formations”.

Line 114. Added the word “Mexico” after “Veracruz State”. 

Line 117. Changed “it is considered a shallow shelf depositional environment with internal neritic conditions and middle neritic influence” to “the environment was interpreted as a shallow marine habitat corresponding to the neritic zone”.

Line 119. Changed “correspond to a reasonably” to “persisted for a prolonged period”

Line 148. Removed the italic font and the translation. However, we prefer to keep the Colección Nacional de Paleontología since it is the specific place in the IGM to accommodate the fossil specimens. 

Lines 149-150. Removed italic font and translation. 

Line 164. Changed “Filo” to “Phylum”. 

Lines 177-178. Changed “Right lateral” to “Distal” and “Left lateral” to “Mesial”. 

Lines 188-190. Changed “depth” to “height”.

Line 192. Changed “lateral” to “in both medial and distal”. The mentioned features can be viewed in both views. 

Line 207. Changed “Copper” to “copper”.

Lines 214-216. Changed “Extant C. brachyurus reaches up to 320 m in marine environments and is generally associated with inshore and reef systems; however, it is reported in estuaries and occasionally in freshwater [37]“ to “Extant C. brachyurus reaches up to 320 m in marine environments and is associated with reef systems, estuaries, and occasionally in freshwater [37]”. 

Line 237-238. Added “to the distal edge”

Line 243. Changed “Cusplet” to “Cusplets”

Line 252. Changed “Bull” to “bull”.

Lines 260-262. Removed “;however”, and added “,and” 

Line 274. Added “(Fig 5 A-D)”.

Line 278. Reference figured. 

Line 281. Every inclined tooth is pointed distally. 

Line 282. Changed “;” to “,”.

Lines 282-283. Changed “lingual is medially concave” to “while the lingual surface is medially concave, forming a sigmoid curvature observed in both distal and mesial views.”

Line 285. Changed “about” to “around”

Line 302. Changed “Hardnose” to “hardnose” 

Lines 306-307. Changed “the lack of the twist in the crow” to “absence of twist in the crown” 

Line 307. Added “found in”.

Line 321. The crown can be upright or curved. The curvature can be to the right or the left side. In all curved crowns, the curvature is directed to the distal portion. In this case we are indicating the direction of the crown curvature. 

Line 332. Removed “in all extensions”.

Line 336. Removed “forming a blade that also is finely serrated.” 

Lines 343-346. The sencentes was changed to “This single specimen is attributed to C. perezii (the Caribbean reef shark) due to the presence of diagnostic features, including a large and triangular crown with vestigial serrations along both the cutting edges and the curvature of the cutting edges not pronounced”.

Line 352. Removed italic from sp.

Line 358. Fig 1 referenced.

Line 361. Changed “unique” to “single”.

Line 361. Changed “depth” to “width”. 

Lines 366-368. Sentence changed.

Lines 369-371. Changed “degrees” to °. Changed along the main text. 

Line 373. Changed to “The shoulders are wide and the enameled portion over its dorsal surface is thin”.

Line 382. Changed Requiem to requiem.

Line 383. Changed shoulder to shoulders.

Line 384. Changed to and a notable transverse groove. 

Line 404. Added (Fig 1). 

Line 407. Changed unique to single.

Lines 411-412. Rewrite as “In both mesial and distal views, the labial surface of the crown is concave, while the lingual is convex.”

Lines 412. Sentences changed to: The apex in Galeocerdo mayumbiensis is directed to the labial portion but does not have the apex sigmoid curvature, such as is found in Carcharhinus sp. for example. 

Lines 421-423. Paragraph changed. 

Lines 427-432. Paragraph changed.

Line 437. Changed

Line 439. Word removed.

Line 450. Word removed.

Line 454. Changed unique to single

Line 455. Changed to wider than long.

Line 462 Changed to snaggletooth

Line 462. Removed the word jaw.

Line 495. Added (Fig 1). 

Line 499. Changed depth to width.

Line 501. Changed crow to crown.

Line 502. Added In both mesial and distal views.

Line 507. Removed the word shallower. 

Line 520. The sentence was changed to “teeth with the crown labio-lingually flattened”. 

Line 521. Changed to well-developed. 

Line 524. The sentence was changed to “The small length indicates that all teeth belonged to small specimens, probably juvenile sharks.”

Line 527. Added: of †Otodus (Carcharocles) megalodon.

Line 537. Added: (Fig 7 Q-T).

Line 544. Changed to: The single tooth is flattened... 

Line 546. Added: in both mesial and distal views. 

Line 548. Changed to “strongly serrate”

Line 555. Changed to The single specimen.

Line 555. Changed to white 

Line 558-560. Sentence changed to: Some hypotheses indicate that these species are not contemporary [39]. Nevertheless, a previous co-occurrence reported in the late Pliocene strata of Baja California [62] and the single species herein described indicates that both species occurred on both sides of the Mexican coasts during the Miocen

---

## [Decision Letter · Decision Letter 1]

10 Oct 2024

PONE-D-24-07932R1Underwater paleontology inside cenotes reveals the Miocene-Pliocene fish diversity in the Yucatan Peninsula, southeast MexicoPLOS ONE

Dear Dr. Cantalice,

Thank you for submitting your manuscript to PLOS ONE. After careful consideration, we feel that it has merit but does not fully meet PLOS ONE’s publication criteria as it currently stands. Therefore, we invite you to submit a revised version of the manuscript that comprehensively addresses the remaining points raised again during the second round of peer review.

We look forward to receiving your revised manuscript.

Kind regards,

Michael Schubert

Academic Editor

PLOS ONE

**Journal Requirements:**

Reviewers' comments:

Reviewer's Responses to Questions

**Comments to the Author**

1. If the authors have adequately addressed your comments raised in a previous round of review and you feel that this manuscript is now acceptable for publication, you may indicate that here to bypass the “Comments to the Author” section, enter your conflict of interest statement in the “Confidential to Editor” section, and submit your "Accept" recommendation.

Reviewer #1: (No Response)

2. Is the manuscript technically sound, and do the data support the conclusions?

Reviewer #1: Partly

3. Has the statistical analysis been performed appropriately and rigorously? 

Reviewer #1: N/A

4. Have the authors made all data underlying the findings in their manuscript fully available?

Reviewer #1: Yes

5. Is the manuscript presented in an intelligible fashion and written in standard English?

Reviewer #1: Yes

6. Review Comments to the Author

**Reviewer #1:** It appears that many of the suggested edits to improve this manuscript by myself and the other reviewer were addressed well, however, authors did not address the nursery hypothesis sufficiently. There is still no comparative discussion about why the locality specimens might be small (e.g., due to ontogeny vs some other size-variability considerations). There is still currently one example for Otodus megeladon, but what else? Because, there are only four specimens of Otodus megalodon, and two of them have cusplets - that cannot indicate predominance, rather only that some of these specimens appear to be juveniles among later growth stages. This section needs revision still - it can be reframed in a comparative way but currently cannot state that this is a predominantly juvenile location, only that specimens are smaller than one would expect.

Lines 828-834: “…compared with other fossil and extant adult species...” From where? This is a valuable discussion point to be more specific with references about other taxa. Small changes in the text to distinguish your work from other previously published work. For instance, do you mean the largest megalodon locally from your study or largest globally ever to exist? In fact, you could answer for both and describe the implications more. Check throughout about consistent change of the term depth. It was done numerous times but not everywhere. Make sure that is intentional, especially because depth is not a term in table S1.

Line, 854. Replace “wholly extinguished” with “extinct”

Line 710: The first sentence is still awkward. Whereas, before I wanted to know what it was poorly understood, I’m still left to wonder and told it needs to be better. I suggest a complete rewrite as, “As a result of this study, the fossil diversity from the Carrillo Puerto Formation is determined to be much greater than previously understood.”

7. PLOS authors have the option to publish the peer review history of their article (what does this mean?). If published, this will include your full peer review and any attached files.

Reviewer #1: No

---

## [Author Response · Author response to Decision Letter 1]

24 Nov 2024

Reviewer #1: It appears that many of the suggested edits to improve this manuscript by myself and the other reviewer were addressed well, however, authors did not address the nursery hypothesis sufficiently. There is still no comparative discussion about why the locality specimens might be small (e.g., due to ontogeny vs some other size-variability considerations). There is still currently one example for Otodus megeladon, but what else? Because, there are only four specimens of Otodus megalodon, and two of them have cusplets - that cannot indicate predominance, rather only that some of these specimens appear to be juveniles among later growth stages. This section needs revision still - it can be reframed in a comparative way but currently cannot state that this is a predominantly juvenile location, only that specimens are smaller than one would expect.

Response: The modifications proposed by the reviewer were also attended to. The hypothesis about the nursery and the presence of juveniles were completely removed, and the manuscript focused only on saying that some specimens in the Carrillo Puerto Formation were small. 

Lines 828-834: “…compared with other fossil and extant adult species...” From where? This is a valuable discussion point to be more specific with references about other taxa. Small changes in the text to distinguish your work from other previously published work. For instance, do you mean the largest megalodon locally from your study or largest globally ever to exist? In fact, you could answer for both and describe the implications more. Check throughout about consistent change of the term depth. It was done numerous times but not everywhere. Make sure that is intentional, especially because depth is not a term in table S1.

Response: The paragraph were modified. And the word depth was revised. 

Line, 854. Replace “wholly extinguished” with “extinct”

Response: modified.

Line 710: The first sentence is still awkward. Whereas, before I wanted to know what it was poorly understood, I’m still left to wonder and told it needs to be better. I suggest a complete rewrite as, “As a result of this study, the fossil diversity from the Carrillo Puerto Formation is determined to be much greater than previously understood.”

Response. Sentence modified.

---

## [Editor Report · Decision Letter 2]

26 Nov 2024

Underwater paleontology inside cenotes reveals the Miocene-Pliocene fish diversity in the Yucatan Peninsula, southeast Mexico

PONE-D-24-07932R2

Dear Dr. Cantalice,

We’re pleased to inform you that your manuscript has been judged scientifically suitable for publication and will be formally accepted for publication once it meets all outstanding technical requirements.

Kind regards,

Michael Schubert

Academic Editor

PLOS ONE

---

## [Editor Report · Acceptance letter]

29 Nov 2024

PONE-D-24-07932R2 

PLOS ONE

Dear Dr. Cantalice, 

I'm pleased to inform you that your manuscript has been deemed suitable for publication in PLOS ONE. Congratulations! Your manuscript is now being handed over to our production team.

Kind regards, 

on behalf of

Dr. Michael Schubert 

Academic Editor

PLOS ONE